



# Tracking vegetation phenology across diverse biomes using Version 3.0 of the PhenoCam Dataset

Adam M. Young[1,2], Thomas Milliman[2,3], Koen Hufkens[4], Keith L. Ballou[5], Christopher Coffey[5], Kai Begay[2], Michael Fell[5,6], Mostafa Javadian[2,6], Alison K. Post[7], Christina Schädel[8], Zakary Vladich[2,6], Oscar Zimmerman[2], Dawn M. Browning[9], Christopher R. Florian[1], Minkyu Moon[10], Michael D. SanClements[1], Bijan Seyednasrollah[2,6], Mark A. Friedl[11], and Andrew D. Richardson[2,6]

[1]National Ecological Observatory Network, Battelle, Boulder, CO, USA
[2]Center for Ecosystem Science and Society, Northern Arizona University, Flagstaff, AZ, USA
[3]Earth Systems Research Center, University of New Hampshire, Durham, NH, USA
[4]BlueGreen Labs (BV), Melsele, Belgium
[5]Information Technology Services, Northern Arizona University, Flagstaff, AZ, USA
[6]School of Informatics, Computing, and Cyber Systems, Northern Arizona University, Flagstaff, AZ, USA
[7]Earth Lab, CIRES, University of Colorado Boulder, Boulder, CO, USA
[8]Woodwell Climate Research Center, Falmouth, MA, USA
[9]USDA ARS, Jornada Experimental Range, Las Cruces, NM, USA
[10]Department of Environmental Science, Kangwon National University, Chuncheon, South Korea
[11]Department of Earth and Environment, Boston University, Boston, MA, USA

*Correspondance to*:

| | | |
|---|---|---|
| Adam M. Young | younga1@battelleecology.org | https://orcid.org/0000-0003-2668-2794 |

*Contact information for co-authors:*

| | | |
|---|---|---|
| Thomas Milliman | thomas.milliman@gmail.com | https://orcid.org/0000-0001-6234-8967 |
| Koen Hufkens | koen.hufkens@gmail.com | https://orcid.org/0000-0002-5070-8109 |
| Keith L. Ballou | keith.ballou@nau.edu | https://orcid.org/0009-0000-4235-7436 |
| Christopher Coffey | chris.coffey@nau.edu | https://orcid.org/0009-0000-4235-7436 |
| Kai Begay | klb792@nau.edu | |
| Michael Fell | michael.fell@nau.edu | https://orcid.org/0000-0001-9713-749X |
| Mostafa Javadian | Mostafa.Javadian@nau.edu | https://orcid.org/0000-0001-7428-8869 |
| Alison K. Post | alison.post@colorado.edu | https://orcid.org/0000-0003-2931-6490 |
| Christina Schädel | cschaedel@woodwellclimate.org | https://orcid.org/0000-0003-2145-6210 |
| Zakary Vladich | zgv4@nau.edu | |
| Oscar Zimmerman | orz6@nau.edu | https://orcid.org/0000-0003-4113-2133 |
| Dawn M. Browning | dawn.browning@usda.gov | https://orcid.org/0000-0002-1252-6013 |
| Christopher R. Florian | cflorian@battelleecology.org | https://orcid.org/0000-0003-4217-0684 |
| Minkyu Moon | moon.minkyu@kangwon.ac.kr | https://orcid.org/0000-0003-0268-1834 |
| Michael D. SanClements | msanclements@battelleecology.org | https://orcid.org/0000-0002-1962-3561 |
| Bijan Seyednasrollah | bijan.s.nasr@gmail.com | https://orcid.org/0000-0002-5195-2074 |
| Mark A. Friedl | friedl@bu.edu | https://orcid.org/0000-0001-6899-2948 |
| Andrew D. Richardson | Andrew.Richardson@nau.edu | https://orcid.org/0000-0002-0148-6714 |





**Abstract.** Vegetation phenology plays a significant role in driving seasonal patterns in land-atmosphere interactions and ecosystem productivity, and is a key factor to consider when modeling or investigating ecological and land-surface dynamics. To integrate phenology in ecological research ultimately requires the application of carefully curated and quality controlled phenological datasets that span multiple years and include a wide range of different ecosystems and plant functional types. By using digital cameras to record images of plant canopies every 30 minutes, pixel-level information from

the visible red-green-blue color channels can be quantified to evaluate canopy greenness (defined as the green chromatic coordinate, $G_{CC}$), and how it varies in space and time. These phenological cameras (i.e., "PhenoCams") offer a pragmatic and effective way to measure and provide phenology data for both research and education. Here, in this dataset descriptor, we present the PhenoCam dataset version 3 (V3.0), providing significant updates relative to prior releases. PhenoCam V3.0 includes 738 unique sites and a total of 4805.5 site years, a 170% increase relative to PhenoCam V2.0 (1783 site years), with

notable expansion of network coverage for evergreen broadleaf forests, understory vegetation, grasslands, wetlands, and agricultural systems. Furthermore, in this updated release, we now include a PhenoCam-based estimate of the normalized difference vegetation index (*cameraNDVI*), calculated from back-to-back visible and visible+near-infrared images acquired from approximately 75% of cameras in the network, which utilize a sliding infrared cut filter. Both $G_{CC}$ and *cameraNDVI* showed similar, but somewhat unique, patterns in canopy greenness and VIS vs. NIR reflectance, across various ecosystems,

indicating their consistent ability to record phenological variability. However, we did find that at most sites, $G_{CC}$ time series had less variability and fewer outliers, representing a smoother signal of canopy greenness and phenology. Overall, PhenoCam greenness as measured by both $G_{CC}$ and *cameraNDVI* provides expanded opportunities for studying phenology and tracking ecological changes, with potential applications to the evaluation of satellite data products, earth system and ecosystem modeling, and understanding phenologically mediated ecosystem processes. The PhenoCam V3.0 data release is

publicly available for download from the Oak Ridge National Lab Distributed Active Archive Center: the source imagery used to derive phenology information is available at https://doi.org/10.3334/ORNLDAAC/2364 (Ballou et al., 2025), and the summarized phenology data are available at https://doi.org/10.3334/ORNLDAAC/2389 (Zimmerman et al., 2025).



# 1 Introduction

The study of vegetation phenology aims to describe and understand the drivers and impacts of reoccurring, seasonal changes in plant growth in terrestrial ecosystems, including periods such as budburst and leaf emergence, fall senescence, and dormancy (Lieth and Radford, 1971; Richardson et al., 2013). Vegetation phenology (hereafter referred to as phenology) is sensitive to variability in temperature and precipitation (Jolly et al., 2005; Rosenzweig et al., 2007; Hufkens et al., 2016; Post et al., 2022), and serves as an indicator of climate change (Schwartz, 1998; Peñuelas et al., 2002). Phenology also exerts
direct influence over dynamics linking the biosphere and atmosphere. For example, inter-annual variation in net ecosystem production is tied to shifts in the timing of green-up and leaf emergence across a range of spatial scales; in North America, warmer temperatures in 2012 resulted in relatively early spring green-up of deciduous forests, and notable increases in annual net ecosystem production for the Eastern US (Wolf et al., 2016). Surface-to-atmosphere latent and sensible heat fluxes are also influenced by phenology, for example, by altering aerodynamic resistance to sensible heat fluxes through
changes in land-surface roughness or by influencing evapotranspiration due to timing of seasonal changes in stomatal conductance (Blanken and Black, 2004; Young et al., 2021). Recent Earth-system modeling experiments have also demonstrated how phenology influences land-atmosphere coupling (Li et al., 2024) and boundary layer height (Li et al., 2023). Understanding the role and drivers of phenology in different ecosystems is important for anticipating future terrestrial ecosystem dynamics that require validated, generalizable phenology modules to be integrated with land-surface and Earth-
system models. While current phenology routines have continued to be improved upon (e.g., Hufkens et al., 2018; Post et al., 2022; Schädel et al., 2023), most models are still unable to capture the full range of variability in phenology patterns observed across a wide range of ecosystems and climate regimes (Li et al., 2022). Continued diagnosis and improvement of phenology models will depend on multi-year records and data products covering broad regional-continental spatial scales of phenology.

Multiple approaches and published data products are currently available for studying phenology. At the global scale, satellite-based remote sensing provides a multi-decadal record of vegetation seasonality, but at coarse spatial resolution. The most widely used metric derived from remote sensing reflectance measurements is the Normalized Difference Vegetation Index ($NDVI$). $NDVI$ is defined as the normalized differences between reflectance values from both visible red ($R$) and near-infrared wavelengths ($NIR$),

$$NDVI = (NIR - R)/(NIR + R), \tag{1}$$

$NDVI$ can be broadly related to vegetation health; during photosynthesis, leaf chlorophyll pigments absorb radiation in the spectrum of visible light, while reflecting radiation in the $NIR$ (Waring and Running, 2007). Time series of $NDVI$ can be used to clearly depict seasonal changes in vegetation activity, and these time series can be further used to identify and extract





phenological transition dates based on the seasonal amplitude of greenness. For example, in deciduous broadleaf forests, the
timing of leaf development and senescence can be estimated when *NDVI* reaches 50% of the total seasonal amplitude.
Products derived from such remote sensing data have been invaluable in advancing our understanding of the role of
phenology in many ecosystems (e.g., Stöckli and Vidale, 2004; Zhang et al., 2013; Jeong et al., 2011). While satellite data
enable global monitoring of phenology, the relatively coarse spatial resolution of most platforms (e.g., 500 m for MODIS)
means individual pixels may contain multiple species, plant functional types, or land-cover types. Furthermore, the temporal
resolution of image acquisition and the multi-day compositing period of many platforms (e.g., 8- and 16-day for MODIS)
result in additional uncertainties, because many phenological transitions can occur within the span of a week (Klosterman et
al., 2014). More recently, satellite products at a higher spatial and temporal resolution have become available (e.g., Moon et
al., 2021), but there remains a tradeoff with the shorter duration of these new data records.

The development and implementation of near-surface remote sensing using digital cameras offers a method for
complementing satellite studies of  vegetation phenology. This approach – commonly referred to as PhenoCam (i.e.,
"Phenological Camera") – uses repeat digital imagery from cameras positioned to overlook ecosystem canopies. Individual
cameras are usually programmed to take multiple images per day (e.g., every 15-30 minutes) (Richardson et al., 2018b).
From digital imagery, digital numbers (*DN*) from the visible red (*R*), green (*G*), and blue (*B*) color channels (i.e., *RGB*) can
be extracted for each pixel. By delineating a region of interest (*ROI*) in the camera field-of-view that directly focuses on the
canopy (or other vegetation of interest), information on vegetation greenness is obtained using a metric of relative greenness
called the green-chromatic coordinate ($G_{CC}$),

$$G_{CC} = \frac{G_{DN}}{G_{DN}+R_{DN}+B_{DN}}, \qquad\qquad (2)$$

Time series summaries of $G_{CC}$ – such as at 1- or 3-day time steps – provide information on how vegetation greenness
changes at a relatively fine temporal scale relative to most satellite-based remote sensing. The PhenoCam approach therefore
directly enhances phenology data derived from satellites: PhenoCams provide phenology data at finer spatial (leaf-to-branch)
and temporal (daily) resolution than is usually possible with satellite based measures, although satellite sensors can provide
much broader spatial coverage (continental-to-global). Previous studies have made extensive use of PhenoCam data to
evaluate satellite phenological data products from MODIS (Klosterman et al., 2014; Richardson et al., 2018a; Liu et al.,
2017), Landsat (Melaas et al., 2016), Harmonized Landsat Sentinel-2 (HLS, Bolton et al., 2020), PlanetScope (Moon et al.,
2021), SPOT-VGT and PROBA-V (Bórnez et al., 2020), VIIRS (Zhang et al., 2018), MERIS (Brown et al., 2017), and
GOES (Wheeler and Dietze, 2021).

The PhenoCam Network (<https://phenocam.nau.edu>) is one of the largest public repositories of phenological  digital
camera imagery and derived data products (Richardson, 2023). The majority of the sites within the PhenoCam network are





located in North America, follow a standardized protocol, and use common hardware (StarDot NetCam SC) that has been
vetted (Sonnentag et al., 2012; Brown et al., 2016; Richardson, 2023). The complementary metal-oxide-semiconductor
(CMOS) imaging sensor within this camera is sensitive to *NIR* wavelengths, and the cut filter used to block wavelengths ≥
700 nm for standard visible-wavelength (*RGB*) imagery is software controlled: with the filter removed, the camera records
an *RGB*+*NIR* image (Petach et al., 2014). The original intent of this design was to enhance photon capture under low-light
conditions and to permit nighttime security monitoring with an infrared illuminator. However, it has also been shown to
offer the potential for the camera to serve as a four-channel imager (red, green, blue, and *NIR*), enabling calculation of a
"camera *NDVI*" from digital numbers and exposure values (Petach et al., 2014) that is similar to the standard *NDVI* metric
calculated using reflectance values from satellite imagery (Eq. 1). To date, the implementation and use of camera *NDVI* from
PhenoCams (hereafter referred to as *cameraNDVI*) has received only minor attention (e.g., Filippa et al., 2018).

In this data descriptor, we introduce the PhenoCam V3.0 public data release, which provides a substantial update to the
V2.0 release (Seyednasrollah et al., 2019), with a 170% increase in total site-years and a better representation of understory
ecosystems, evergreen broadleaf forests, grasslands, wetlands, and agriculture systems, in particular. In this descriptor for
the V3.0 dataset, we detail how the PhenoCam Network has grown in terms of spatial and temporal coverage, while also
evaluating the representation of the Network across ecoregions and biomes, at both continental and global levels.
Furthermore, two new operational data records are introduced to enhance the usefulness of this dataset. First, the dataset now
includes *cameraNDVI* (Data Record 6) for all sites with the requisite hardware and camera configuration. We evaluate this
*cameraNDVI* product in a detailed comparison using PhenoCam imagery and *NDVI* estimates derived from broadband
measurements of incident and upwelling solar radiation (i.e., *broadbandNDVI*) (Huemmrich et al., 1999; Jenkins et al.,
2007). We conduct this evaluation using data from National Ecological Observatory Network (NEON;
http://www.neonscience.org/sites; Metzger et al., 2019), spanning a broad range of ecosystems, from Arctic tundra to
tropical forests. Second, we now also include a reduced set of simplified data products containing just three columns: date,
mean measured $G_{CC}$ at a daily time step, and a smoothed $G_{CC}$ product that can be used for interpolation or gap filling (Data
Record 7). For many users, the simplified data will be much easier to work with than the 1- and 3-day summary products
contained in Data Record 4, which are almost 50 columns wide. While it is well established that PhenoCams are a powerful
tool to monitor trends in Phenology, other potential applications of PhenoCam data include: (1) evaluation of satellite data
products; (2) calibration and validation of phenological models for different vegetation types; and (3) ecological
interpretation of other data streams, including eddy covariance data for surface-atmosphere $CO_2$, $H_2O$, and sensible-heat
fluxes.

The data described here have been archived with the ORNL DAAC (Zimmerman et al., 2025) and are also accessible
through the PhenoCam Explorer web page (<https://phenocam.nau.edu/phenocam_explorer>). The data records have been
truncated at the end of 2023, but data records from active cameras continue to be updated nightly, and are publicly available



as provisional (i.e., uncurated) data through the PhenoCam project web page (<https://phenocam.nau.edu>). A companion data set (Ballou et al., 2025), which contains the imagery from which these data are derived, is also being released at the same time, and it may be useful for computer vision, machine learning, or deep learning analyses (e.g., Taylor and Browning, 2022; Cao et al., 2021). See Sect. 5 for our Data Availability statement.



## 2    Methods and materials

The details of camera installation and configuration protocols, site classification, and image and data processing routines have been previously documented in detail by Richardson et al. (2018b) and Seyednasrollah (2019). We provide only a brief summary here, as the underlying methods and data processing code remain unchanged.

### 2.1 Overview of PhenoCam

Each PhenoCam camera is classified into one of three classes: Type I, Type II or Type III. Type I cameras follow a standardized protocol, and site personnel are actively engaged as PhenoCam collaborators (e.g. providing camera maintenance and troubleshooting as required). For Type II cameras, there is some deviation from the standard protocol, but site personnel are still actively engaged. For Type III cameras, there is some deviation from the standard protocol, and no active collaboration of personnel on-site. Because the standard protocol has been widely embraced by PhenoCam network

collaborators (as of 12 December 2024, 836 of 977 cameras with data in the archive, or almost 86%, are classified as Type I), and because of the generally lower data quality from Type III cameras (e.g., issues with white balance, field of view shifts, and interrupted continuity), recent data curation efforts have focused on Type I cameras, and have been discontinued for Type III cameras.

All cameras in the PhenoCam network record three-layer JPEG images, from which we extract information about the

mean intensity of each of the red, green, and blue (RGB) color channels, calculated across a user-defined region of interest (*ROI*), as described in the Introduction (Section 1; e.g., Eq. 2). The *ROI* is delineated to correspond to the vegetation under study (Sonnentag et al., 2012; Richardson et al., 2018b). While a single image per day would be generally sufficient to document phenological changes in most ecosystems, it is typical for cameras in the PhenoCam network to upload an image every 15 or 30 minutes. This ensures high quality data by minimizing data discontinuity in cases of unfavorable weather

(rain or snow), adverse illumination conditions (clouds or aerosols), or short-term power outages. Following previously developed methods (Sonnentag et al., 2012), we use statistics calculated from the sub-daily $G_{CC}$ time-series to generate 1-day and 3-day "summary product" $G_{CC}$ time-series, which have been found to be effective at filtering out noisy color output due to adverse conditions that may occur (Sonnentag et al., 2012). From these summary time series products, we estimate phenological transition dates corresponding to the start of each "greenness rising" (e.g., budburst) and "greenness falling"

phenological phase (e.g., senescence). Uncertainties are quantified and provided for all $G_{CC}$ time series and transition date estimates.

### 2.2 *NDVI* derived from infrared PhenoCam imagery



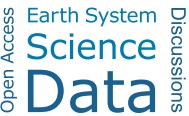

Motivation and proof-of-concept for *cameraNDVI*, as well as details on the calculations, can be found in Petach et al. (2014). In brief, *cameraNDVI* is calculated using data extracted from the same *ROI* in back-to-back (30 seconds apart) *RGB* and *RGB+NIR* images. Accounting for exposure differences between the two images, it is then possible to estimate the *NIR* contribution to the *RGB+NIR* image by subtracting off the estimated *RGB* component

$$NIR_{DN'} = (RGB + NIR)_{DN'} - RGB_{DN'},$$ (3)

Where primes (′) denote exposure-adjusted *DN* values, with $NIR_{DN'}$ and $R_{DN'}$ then used in Eq. 1 in place of reflectances to obtain *cameraNDVI*. We describe some important distinctions between *cameraNDVI* and *NDVI* estimated from other platforms (e.g., satellite remote sensing, or tower-mounted radiometric instruments) in the Discussion (Sect. 4).

### 2.3 Comparisons among *cameraNDVI*, $G_{CC}$, and tower broadband *NDVI*

To investigate how well time series of *cameraNDVI* agree with other estimates of plant phenology, we compared *cameraNDVI* to both $G_{CC}$ (Eq. 2) and tower-measured *broadbandNDVI* (Jenkins et al., 2007). First, to compare *cameraNDVI* and $G_{CC}$, we began with visual comparisons between a sample set of time series to evaluate overall coherence, subsequently calculating and comparing the signal-to-noise ratio (*SNR*) for *cameraNDVI* and $G_{CC}$ at all sites. Specifically, the signal of a given time series (i.e., either $G_{CC}$ or *cameraNDVI*) is characterized using the same smoothing spline approach used to derive seasonal transition dates (Richardson et al., 2018b; Seyednasrollah et al., 2019), where the optimal span of the function is determined by minimizing the Bayesian Information Criterion (Richardson et al., 2018b). The noise is characterized from the residuals around the smoothing spline, and the unitless Signal-to-Noise ratio (*SNR*) is then calculated as the ratio of the variance of the smoothing spline to the variance of the residuals. The *SNR* thus provides a normalized metric of the day-to-day variation in a time series relative to the seasonal variability in that time series. Next, we calculated the ratio of the *SNR* for $G_{CC}$ to the *SNR* for *cameraNDVI* by site when both metrics were available. For interpretation, if $SNR(G_{CC})/SNR(cameraNDVI) > 1$, then $G_{CC}$ is inferred as a "less noisy" index relative to *cameraNDVI*.

In addition to comparing *cameraNDVI* to $G_{CC}$, we further compared *cameraNDVI* to a vegetation index commonly referred to as "broadband *NDVI*" (e.g., Jenkins et al., 2007). Broadband *NDVI* ( hereafter *broadbandNDVI*) is calculated from radiometric sensors that measure downwelling (↓) and upwelling (↑) photosynthetically active radiation ($Q$, 400-700 nm) – measured using photosynthetic photon flux density ($\mu$mol m$^{-2}$ s$^{-1}$) – and global radiation ($Rg$, 400-2500 nm, W m$^{-2}$), where an estimate of reflectances ($r$) are obtained following Jenkins et al. (2007):

$$r_{Tot} = \frac{Rg_{\uparrow}}{Rg_{\downarrow}},$$ (4)



$$r_{VIS} = \frac{Q_\uparrow}{Q_\downarrow}, \tag{5}$$

$$r_{NIR} = 2 \times r_{Tot} - r_{VIS}, \tag{6}$$

$$broadbandNDVI = \frac{r_{NIR} - r_{VIS}}{r_{NIR} + r_{VIS}}, \tag{7}$$

It should also be noted that there are multiple approaches to calculating *broadbandNDVI*, and the calculated index value will vary slightly depending on the approach (e.g., Huemmrich et al., 1999; Wang et al., 2004; Jenkins et al., 2007; Rocha et al.,
2021).

We chose to compare *cameraNDVI* to *broadbandNDVI,* rather than to satellite-based *NDVI,* for several reasons. First, similar to *cameraNDVI*, *broadbandNDVI* estimates are inherently less sensitive to days with clouds, rain, or other adverse weather conditions; they thus have a temporal resolution and continuity that is more closely aligned with *cameraNDVI*. Second, the coarse spatial resolution of many satellite data products risks mixing vegetation types in heterogenous terrain
(Richardson et al., 2018a), while the limited temporal resolution presents further challenges for characterizing the congruency between *cameraNDVI* and satellite *NDVI*. These issues are minimized for *broadbandNDVI* measurements obtained from the same tower where PhenoCams are mounted.

Similar to our comparisons between $G_{CC}$ and *cameraNDVI*, we also compared *cameraNDVI* to *broadbandNDVI* through a simple visual evaluation of the two time series, as well as the same *SNR* analysis. For these comparisons, we used data
aggregated to a 3-day time step. Given the large number of sites for which *broadbandNDVI* can be calculated (183 AmeriFlux sites as of March 11, 2024; <https://ameriflux.lbl.gov/sites/site-search/#vars=PPFD_IN%2CPPFD_OUT%2CSW_IN%2CSW_OUT>) we chose to focus our attention on sites within the National Ecological Observatory Network (NEON, <https://www.neonscience.org>). NEON instruments are rigorously calibrated, and consistent deployment protocols ensure data are comparable across a wide range of site conditions. Across all
NEON sites, a Hukseflux NR01 four-channel net radiometer is deployed to calculate upwelling and downwelling shortwave and longwave radiation (National Ecological Observatory Network, 2023a) and a Kipp & Zonen PAR Quantum Sensor (PQS) 1 was used to measure incoming and outgoing photosynthetically active radition (National Ecological Observatory Network, 2023b). Both these NEON data products (PAR and radiation data) were downloaded from the AmeriFlux data portal for all 47 terrestrial sites in NEON's 20 ecoclimatic domains under the AmeriFlux CC-BY-4.0 License
(<https://ameriflux.lbl.gov/data/data-policy/#cc-by-4>). DOI citations for these downloads are available Table S1. Furthermore, the calculated *broadbandNDVI* data in comparison to *cameraNDVI* are available either in Fig. 6 or in the Supplementary Information (Figs. S2-S6). Finally, all PhenoCam derived variables (e.g., $G_{CC}$, *cameraNDVI*) at the 47



NEON sites were derived from cameras and imagery maintained and operated by NEON (National Ecological Observatory Network, 2023c).

## 2.4 Structure and availability of PhenoCam V3.0 data product

The PhenoCam Dataset V3.0 contains seven separate Data Records for each site (Box 1). Data Records 1-5 are described in detail in Richardson et al. (2018b) and Seyednasrollah et al. (2019); that information is not repeated here. The architecture of the V3.0 dataset is similar to that used in V1.0 and V2.0, and the format of key data files in Data Records 4 and 5 is unchanged to facilitate interoperability with existing data analysis packages such as *phenocamr* (Hufkens et al., 2018). Data Records 6 and 7 are new to this release:

1. New Data Record 6 includes derived data and metadata used to calculate *cameraNDVI*. There are two key file types here:
   a) the "PhenoCam Camera NDVI ROI (RGB/IR Image Pair) Statistics File" (filename: <sitename>_<veg_type>_<ROI_ID_number>_ndvi_roistats.csv) (see Box 2a for details)
   b) the "PhenoCam 1-day and 3-day NDVI Summary Files" (filename: <site_name>_<vegetation_type>_<ROI>_ndvi_1day.csv or _3day.csv) (Box 2b). Note that transition dates are not calculated from the *cameraNDVI* time series.

2. New Data Record 7 provides a set of "simplified" data products, which do not include all the color statistics, color indices, cross-correlations, and uncertainties for different temporal resolutions and filtering approaches that are provided in Data Records 3-5. Rather, Data Record 7 only includes a summary file of daily mean $G_{CC}$ and smoothed daily mean $G_{CC}$ (filename: <site_name>_<vegetation_type>_<ROI>_simplified_1day.csv) (Box 3a), as well as "rising" and "falling" transition dates derived from the daily mean $G_{CC}$ data (filename: <site_name>_<vegetation_type>_<ROI>_simplified_transition_dates.csv) (Box 3b). While these data records were developed with secondary and post-secondary educational applications in mind, we anticipate that most users of the data set will find the simplified data products are sufficient for most scientific applications, with the added benefit of being more compact and easier to work with.





<sitename>_<veg_type>_<ROI_ID_number>
└ data_record_1 (contains general metadata for each site)
- <sitename>_meta.json
- <sitename>_meta.txt

└ data_record_2 (contains the ROI list files and image mask files used for image processing)
- <sitename>_<veg_type>_<ROI_ID_number>_roi.csv
- <sitename>_<veg_type>_<ROI_ID_number>_<mask_index>.tif

└ data_record_3 (contains all-image time series of ROI color statistics based on RGB channels, calculated for every image in the archive, using data_record_2)
- <sitename>_<veg_type>_<ROI_ID_number>_roistats.csv

└ data_record_4 (contains summary time series of ROI color statistics, calculated for 1 and 3 day aggregation periods from data_record_3)
- <sitename>_<veg_type>_<ROI_ID_number>_1day.csv
- <sitename>_<veg_type>_<ROI_ID_number>_3day.csv

└ data_record_5 (contains phenological transition dates, calculated from Gcc in data_record_4)
- <sitename>_<veg_type>_<ROI_ID_number>_1day_transition_dates.csv
- <sitename>_<veg_type>_<ROI_ID_number>_3day_transition_dates.csv

└ data_record_6 (contains ROI statistics for paired RGB-IR images, as used to calculate camera NDVI, as well as 1 and 3 day summary time series for camera NDVI. Note that phenological transition dates are not calculated for NDVI)
- <sitename>_<veg_type>_<ROI_ID_number>_ndvi_roistats.csv
- <sitename>_<veg_type>_<ROI_ID_number>_ndvi_1day.csv
- <sitename>_<veg_type>_<ROI_ID_number>_ndvi_3day.csv

└ data_record_7 (simplified data files)
- <sitename>_<veg_type>_<ROI_ID_number>_simplified_1day.csv
- <sitename>_<veg_type>_<ROI_ID_number>_simplified_transition_dates.csv

**Box 1.** Dataset hierarchy of PhenoCam V3.0. Each ROI for each site has 7 data structures, with each structure representing a different level of processing. For data downloaded from the PhenoCam Explorer web page, data for each <sitename>_<veg_type>_<ROI_ID_number> will be contained in a single .zip file, with each data record in a separate folder.

270



**An example of a "Camera NDVI ROI Statistics File" from Data Record 6 (for display purposes the lines have been broken with a '\' character):**

```
#
#
# NDVI statistics timeseries for alligatorriver
#
# Site: alligatorriver
# Veg Type: DB
# ROI ID Number: 1000
# Lat: 35.7879
# Lon: -75.9038
# Elev: 1
# UTC Offset: -5
# Resize Flag: False
# Version: 1
# Creation Date: 2021-12-03
# Creation Time: 10:00:17
# Update Date: 2021-12-03
# Update Time: 10:00:17
#
date,local_std_time,doy,filename_rgb,filename_ir,solar_elev,exposure_rgb,exposure_ir,mask_index,\
r_mean,g_mean,b_mean,ir_mean,ir_std,ir_5_qtl,ir_10_qtl,ir_25_qtl,ir_50_qtl,ir_75_qtl,ir_90_qtl,ir_95_qtl,gcc,\
Y,Z_prime,R_prime,Y_prime,X_prime,NDVI_c
2012-05-06,07:31:09,127,alligatorriver_2012_05_06_073109.jpg,alligatorriver_IR_2012_05_06_073032.jpg,27.9754,355,67,1,\
91,107,54,96,20.5030,61.0000,70.0000,84.0000,99.0000,111.0000,122.0000,128.0000,0.4226,\
96.7480,11.8496,4.8755,5.1348,6.7147,0.1587
2012-05-06,08:01:09,127,alligatorriver_2012_05_06_080109.jpg,alligatorriver_IR_2012_05_06_080031.jpg,34.0538,224,40,1,\
96,107,48,99,21.2579,62.0000,72.0000,87.0000,101.0000,114.0000,126.0000,132.0000,0.4255,\
97.7713,15.7978,6.4400,6.5326,9.2652,0.1799
2012-05-06,08:31:09,127,alligatorriver_2012_05_06_083109.jpg,alligatorriver_IR_2012_05_06_083031.jpg,40.1044,148,18,1,\
92,104,49,96,25.1568,57.0000,67.0000,81.0000,96.0000,112.0000,130.0000,141.0000,0.4246,\
94.6368,22.8614,7.5662,7.7791,15.0823,0.3319
...
```

Comment lines at the beginning of the file are preceeded with '# ' and include some basic site metadata along with creation and update dates and times. (The long lines have been broken up here with a '\' character for display purposes). The columns in the file are:

- **date**: local date for image
- **local_std_time**: local standard time
- **doy**: day of year
- **filename_rgb**: RGB filename
- **filename_ir**: IR filename
- **solar_elev**: solar elevation angle
- **exposure_rgb**: exposure of RGB image
- **exposure_ir**: exposure of IR image
- **mask_index**: index into mask list
- **r_mean**: mean red digital number (DN) over the ROI
- **g_mean**: mean green digital number (DN) over the ROI
- **b_mean**: mean blue digital number (DN) over the ROI
- **ir_mean**: mean digital number (DN) over the ROI from the IR image
- **ir_std**: standard deviation of digital number (DN) over the ROI from the IR image
- **ir_5_qtl,ir_10_qtl,ir_25_qtl,ir_50_qtl,ir_75_qtl,ir_90_qtl,ir_95_qtl**: the 5,10,...,90,95 quantile values of the DN values over the ROI
- **gcc: gcc calculated across the ROI, from the RGB image**
- **Y,Z_prime,R_prime,Y_prime,X_prime**: intermediate values for camera NDVI calculation
- **NDVI_c**: camera NDVI as calculated in Petach et al. (2014)



**Box 2a. Format of "Camera NDVI ROI (RGB/IR Image Pair) Statistics File" in Data Record 6**: The Camera NDVI ROI statistics
file (filename: <sitename>_<veg_type>_<ROI_ID_number>_ndvi_roistats.csv) is created by combining the RGB and IR ROI statistics
files for RGB/IR image pairs.



An example of a "**1-day Camera NDVI Summary File"** from Data Record 6 (for display purposes the lines have been broken with a '\' character). The format of the 3-day file is identical; only the aggregation period changes.

```
#
# 1-day NDVI summary timeseries for coweeta
#
# Site: coweeta
# Veg Type: DB
# ROI ID Number: 2000
# Lat: 35.0592
# Lon: -83.4275
# Elev: 680
# UTC Offset: -5
# Image Count Threshold: 1
# Aggregation Period: 1
# Solar Elevation Min: 10.0
# Time of Day Min: 00:00:00
# Time of Day Max: 23:59:59
# ROI Brightness Min: 100
# ROI Brightness Max: 665
# Creation Date: 2021-12-03
# Creation Time: 11:52:22
# Update Date: 2021-12-03
# Update Time: 11:52:24
#
date,year,doy,image_count,midday_rgb_filename,midday_ir_filename,midday_ndvi,gcc_90,ndvi_mean,ndvi_std,\
ndvi_50,ndvi_75,ndvi_90,max_solar_elev,snow_flag,outlierflag_ndvi_mean,outlierflag_ndvi_50,\
outlierflag_ndvi_75,outlierflag_ndvi_90
2016-06-22,2016,174,25,coweeta_2016_06_22_115306.jpg,coweeta_IR_2016_06_22_115306.jpg,0.04350,0.43888,\
0.38738,0.10207,0.37990,0.40910,0.53376,78.04090,NA,NA,NA,NA,NA
2016-06-23,2016,175,26,coweeta_2016_06_23_115306.jpg,coweeta_IR_2016_06_23_115306.jpg,0.66230,0.42720,\
0.35763,0.09380,0.34935,0.39308,0.43970,78.01470,NA,NA,NA,NA,NA
2016-06-24,2016,176,26,coweeta_2016_06_24_115305.jpg,coweeta_IR_2016_06_24_115305.jpg,0.38210,0.42780,\
0.32801,0.11414,0.35570,0.39028,0.40970,77.98180,NA,NA,NA,NA,NA
...
```

Comment lines at the beginning of the file are preceeded with '# ' and include some basic site metadata along with creation and update dates and times. Dates for which there are no images (or none passing the selection criteria) have empty fields as show in the second data line above. When a particular value cannot be calculated it is given a "no data" value of NA. The columns in the file are:

- **date**: local date of middle of time period (1-day or 3-day)
- **doy**: doy for this date. The date/doy values chosen are for fixed days-of-year.
  (For the 3-day summary file these will be doy=2, 5, 8, etc.)
- **image_count**: number of images passing the selection criteria
- **midday_rgb_filename**: filename for the RGB image which is closest to noon (midday image) on the middle day of summary period
- **midday_ir_filename**: filename for the IR image which is closest to noon (midday image) on the middle day of summary period
- **midday_ndvi**: mean NDVI DN over ROI for the midday image
- **gcc_90**: 90th percentile gcc value for all the image pairs passing the selection criteria
- **ndvi_mean**: mean NDVI value for all the image pairs passing the selection criteria
- **ndvi_std**: standard deviation of NDVI values for all the image pairs passing the selection criteria
- **ndvi_50, ndvi_75, ndvi_90**: 50th, 75th, and 90th percentiles of NDVI values
- **max_solar_elev**: maximum solar elevation for the images from this day
- **snow_flag**: snow flag (1=snow present, 0=snow NOT present))
- **outlierflag_ndvi_mean**: outlier flag for NDVI mean value (1=outlier) [note: at present, outlier flags are not being calculated for NDVI]
- **outlierflag_ndvi_50**: outlier flag for NDVI 50th percentile value (1=outlier)
- **outlierflag_ndvi_75**: outlier flag for NDVI 75th percentile value (1=outlier)
- **outlierflag_ndvi_90**: outlier flag for NDVI 90th percentile value (1=outlier)



**Box 2b. Format of the "1-day and 3-day NDVI Summary Files" in Data Record 6.** Derived from the "Camera NDVI ROI (RGB/IR Image Pair) Statistics File", this file (filename: <sitename>_<veg_type>_<ROI_ID_number>_ndvi_1day.csv or _3day.csv) reports aggregated statistics for $G_{cc}$ and *camera NDVI* calculated over 1- and 3-day aggregation periods.





**The "Simplified Daily Summary Files" from Data Record 7** are intended to be easier for data end-users to work with, in that they do not have the multitide of columns found in Data Records 3 and 4. Additionally, unlike the other standard data records, the simplified data records do not include any metadata. Here is an an example of one of these flat-text, comma-delimited files:

```
date,gcc_mean,smooth_gcc_mean
... [filled lines omitted],,
4/6/08,0.3526,0.353
4/7/08,0.3606,0.3544
4/8/08,0.3627,0.356
4/9/08,0.3632,0.3574
4/10/08,0.3615,0.3586
...
```

The columns in the file are:
- **date**: local date
- **gcc_mean**: mean daily $G_{cc}$ value, from data record 4
- **smooth_gcc_mean**: smoothed value of $G_{CC}$ from the optimized spline, from data record 4

**Box 3a. Format of the "Simplified Daily Summary Files" in Data Record 7.** This file (filename:
<sitename>_<veg_type>_<ROI_ID_number>_simplified_1day.csv) reports aggregated statistics for $G_{cc}$_mean at a 1-day aggregation period.

The "**Simplified Transition Date Files**" from Data Record 7 include only transition dates derived from $G_{cc}$_mean. This file is intended to be easier for data end-users to work with, compared to the standard transition date files in Data Record 5 which also include information about uncertainties and the seasonal amplitude of $G_{cc}$. Additionally, the simplified data records do not include any metadata. Here is an example of one of these flat-text, comma-delimited files:

```
year,direction,date_10,date_25,date_50,DOY_10,DOY_25,DOY_50
2008,rising,1-May,7-May,14-May,122,128,135
2008,falling,22-Oct,18-Oct,12-Oct,296,292,286
2009,rising,24-Apr,1-May,10-May,114,121,130
2009,falling,21-Oct,17-Oct,9-Oct,294,290,282
2010,rising,17-Apr,25-Apr,3-May,107,115,123
...
```

The columns in the file are:
- **year**: year in which the transition occurred.
- **direction**: indicates whether the reported transition dates correspond to a "greenness rising" or "greenness falling" stage. Note that there may be more than one rising/falling cycle per calendar year, and a single rising or falling stage may cut across years.
- **transition_10, transition_25, transition_50**: the extracted transition dates (format MM-DD) for each "greenness rising" or "greenness falling" stage, corresponding to 10%, 25% and 50% of the $G_{CC}$ amplitude of that stage.
- **date_10, date_25, date_50**: day-of-year values corresponding to the calendar date transitions reported in the previous three columns

**Box 3b. Format of the "Simplified Transition Date Files" in Data Record 7.** This file (filename:
<sitename>_<veg_type>_<ROI_ID_number>_simplified_transition_dates.csv) reports transition dates for $G_{cc}$_mean, extracted from the 1-day transition dates reported in Data Record 5.





## 3 Results

### 3.1 Updated Data Coverage of V3.0

The PhenoCam V3.0 dataset release has significantly expanded in both spatial and temporal coverage relative to PhenoCam V2.0 (Fig. 1, Table 1). Sites included in this data release have at minimum six months of continuous data available, and all time series have been carefully curated via repeated visual evaluations and quality checks by an expert team. Adjustment of ROI masks have been made as needed to accommodate camera field of view shifts, and Type II or III sites where automatic white balancing has negatively affected data quality have been removed. There are now 738 unique sites and 4805.5 site-

years within this data release, compared to 393 sites and 1783 site-years in PhenoCam V2.0 (Seyednasrollah et al. 2019) (Table 1). The vegetation types with the largest increase in site-years (as a percentage) were:

1. 1118% increase for understory (UN), from 18 sites-years in V2.0 to 219.2 site-years in V3.0.
2. 264% increase for evergreen broadleaf forests (EB), from 28 sites-years in V2.0 to 101.8 site-years in V3.0
3. 227% increase in grasslands (GR), from 279 to site-years in V2.0 to 912.4 site-years in V3.0.

4. 217% increase in wetlands (WL), from 142 site-years in V2.0 to 436.8 site-years in V3.0.

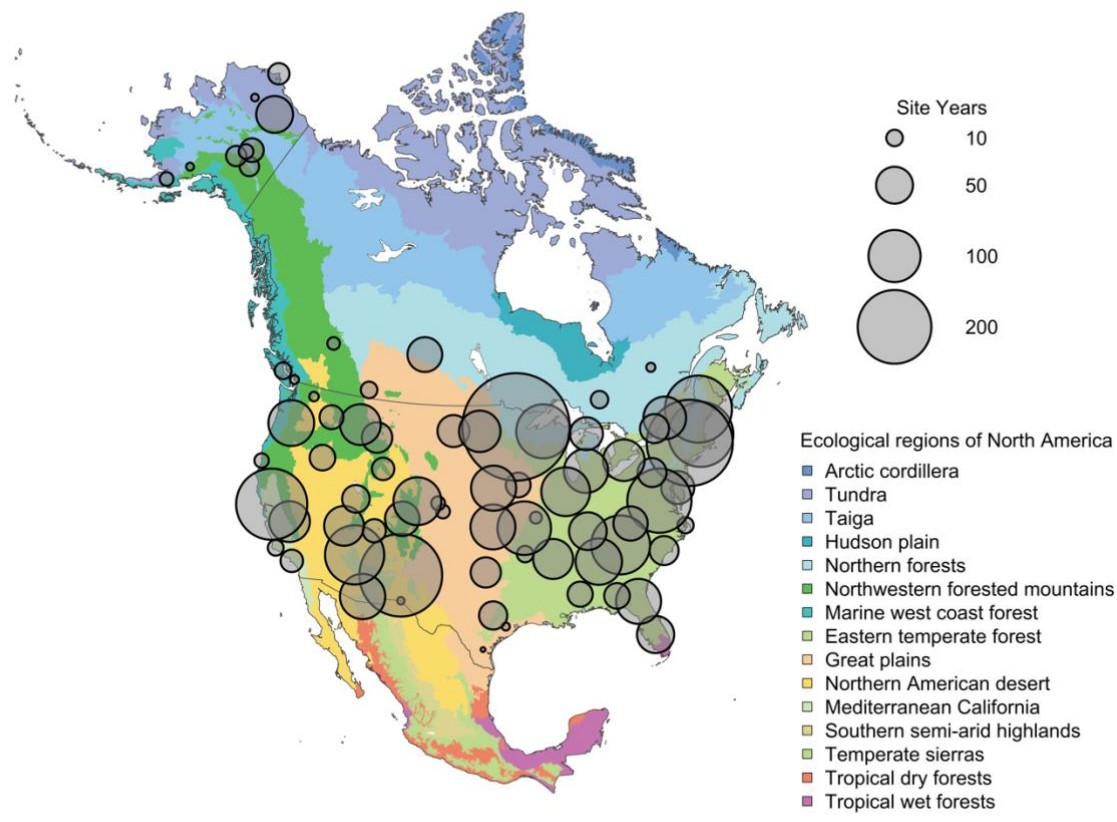

**Figure 1. Spatial distribution of PhenoCam data across ecological regions of North America.** Background map illustrates USA Environmental Protection Agency Level I Ecoregions (Omernik and Griffith, 2014). Data counts have been aggregated to a spatial resolution of 4°, and the size of each circle corresponds to the number of site-years of data in the 4x4° grid cell. A total of 4286.6 out of 4805.5 total site years in the V3.0 dataset are depicted in Fig. 1. However, sites in Hawaii, Puerto Rico, Central and South America, Europe, Asia and Africa (total of 518.9 site years) are not shown.






**Table 1. Vegetation type abbreviations for ROIs (region of interests), and the corresponding number of site-years of data in the PhenoCam dataset described here (V3.0)**. For comparative purposes, the number of site-years of data in the previous dataset releases is also presented. MX and NV ROIs were excluded in V2.0 but are currently available again in V3.0. Please note there are 2.7 site years of Reference Panel (RF) ROIs in V3.0 as well, for a total of 4805.5 site years in the V3.0 data release.


| Abbreviation | Description | Site-years in Dataset V1.0 | Site-years in Dataset V2.0 | Site-years in Dataset V3.0 |
|---|---|---|---|---|
| AG | Agriculture | 50 | 226 | 703.5 |
| DB | Deciduous Broadleaf | 392 | 653 | 1185.2 |
| DN | Deciduous Needleleaf | 4 | 45 | 115.3 |
| EB | Evergreen Broadleaf | 2 | 28 | 101.8 |
| EN | Evergreen Needleleaf | 80 | 264 | 778.0 |
| GR | Grassland | 121 | 280 | 912.4 |
| MX | Mixed vegetation (generally EN/DN, DB/EN, or DB/EB) | 5 | - | 13.7 |
| NV | Non-vegetated | 14 | - | 17.2 |
| SH | Shrubs | 46 | 141 | 436.8 |
| TN | Tundra (includes sedges, lichens, mosses, etc.) | 22 | 68 | 117.0 |
| UN | Understory | - | 18 | 219.2 |
| WL | Wetland | 11 | 58 | 202.7 |





Using the Level II Ecoregion classification of North America (<https://www.epa.gov/eco-research/ecoregions-north-america>), we identified ecoregions and biomes where coverage is lowest. From about 30°N to 55°N, virtually every Level

II ecoregion has at least three (and in many cases substantially more) PhenoCams (Figure 2a). Ecoregions in the high Arctic of northern Canada and most of Mexico emerge as poorly represented, suggesting they should be targeted for future camera deployment efforts. The *everglades* ecoregion of Southern Florida does not have any PhenoCams currently, but there are six active PhenoCams in Puerto Rico to characterize coverage of North American tropical wet forests. Using the Whittaker Biome Classification (Whittaker, 1975), we also examined the distribution of PhenoCam sites across global climate-space

(Figure 2b). Using the most recent version of WorldClim climatological temperature and precipitation data (Fick and Hijmans, 2017), we found that mean annual temperature at PhenoCam sites in North America spans almost 40°C, ranging from -12.0°C to 26.1°C, while mean annual precipitation varies 30-fold, from 109 mm to over 3800 mm. Among the biomes corresponding to this climatic gradient, boreal forest, temperate forest, temperate grassland desert, temperate rain forest, tropical forest savanna, and woodland/shrubland biomes are generally well-represented by the current distribution of

PhenoCam network sites. However, the climate representation of the network would benefit from the installation of more cameras in subtropical desert, tundra, and tropical rain forest biomes. Although expansion of PhenoCam coverage in Mexico is expected in the coming years, increased global coverage of warm, wet, and warm and wet ecosystems will require collaboration and engagement of site PIs across the tropics and sub-tropics more generally.





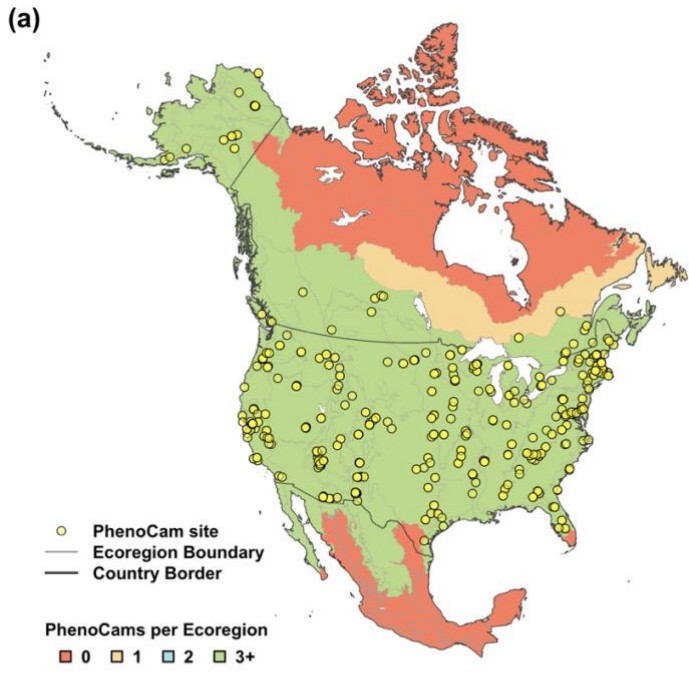

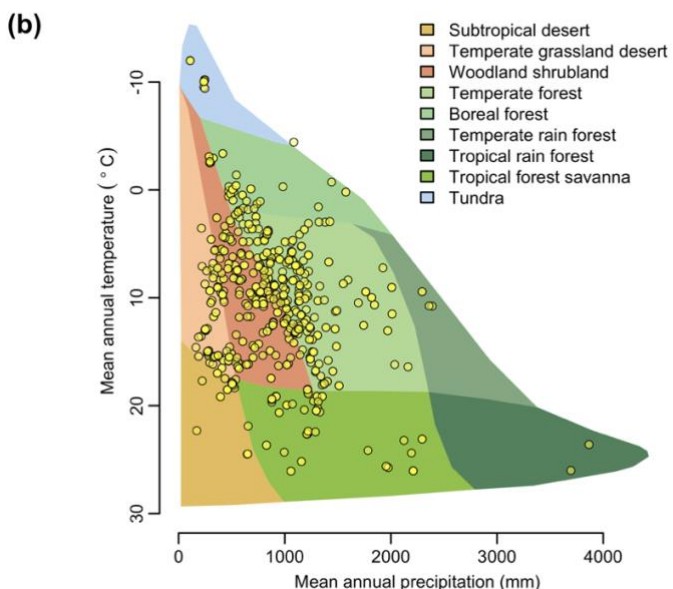

**Figure 2. Representation of PhenoCam cameras in both geographical and climatological space.** (**a**) The number of PhenoCams for each Level II Ecoregion in North America. colored by the number of PhenoCams per region. (**b**) The distribution of PhenoCams across climate space in relation to major terrestrial biomes as well defined by the Whittaker classification. Ecoregions boundaries are obtained from the USA Environmental Protection Agency Ecoregion Level II map of North America from Omernik and Griffith (2014).



The longest time series for a single plant functional type (PFT) and Type I camera at a single site is for a evergreen
conifer forest site, *howland1,* started in January 2007 (16.8 yr split into three separate ROIs that adjust for camera field-of-
view shifts; <https://phenocam.nau.edu/webcam/sites/howland1/>). Other Type I cameras of considerable temporal coverage
include four deciduous broadleaf forest sites where cameras were first installed in 2008: *harvard* (15.7 yr), *caryinstitute*
(15.7 yr), *queens* (15.5 yr), *bartlettir* (15.4 yr), and *morganmonroe* (15.3 yr). In total, there are 51 time series from Type I
cameras that are at least a decade in length, and 355 Type I sites with time series for a single PFT between 5-10 years in
length. Of cameras with the capacity to produce *cameraNDVI*, the longest ROIs are more than 10 years long (e.g.,
*canadaOBS*, *kendall*, *missouriozarks*), with 341 ROIs at least 5 years in length.

### 3.2   Comparisons among $G_{CC}$, *cameraNDVI*, and *broadbandNDVI*

We generally found that $G_{CC}$ and *cameraNDVI* exhibited very similar patterns in canopy greenness (Fig. 3), indicating the
capacity of both $G_{CC}$ and *cameraNDVI* to consistently record variability in phenology. This similarity was apparent across a
wide range of ecosystems, from Arctic tundra to deciduous forest ecosystems, as well as shrublands and grasslands. While
there was general agreement in seasonal patterns, there were some distinct and important differences as well. As an example,
there were several key discrepancies between $G_{CC}$ and *cameraNDVI* at deciduous broadleaf sites (Figs. 3-4). First, there is no
distinct "spike" in spring greenness in early spring in *cameraNDVI*, a common and notable artefact in $G_{CC}$ caused by bright
"greenness" of early season leaves (Keenan et al., 2014) (Fig. 4). Additionally, there is a delay in fall senescence in
*cameraNDVI* relative to $G_{CC}$, with *cameraNDVI* exhibiting a more gradual decline in greenness after October; this is
presumed to be driven by differences in foliage color (affecting $G_{CC}$) vs. foliage amount (affecting *cameraNDVI*). For both
of these reasons, *cameraNDVI* likely better represents the seasonal dynamics of deciduous forest LAI (leaf area index); but,
because $G_{cc}$ and *cameraNDVI* are indicative of different aspects of phenology (leaf color vs. leaf presence), we believe that
the "best" metric will depend on the specific application. In this sense, the two metrics are complementary rather than
redundant.







**Figure 3. Time series comparing *cameraNDVI* to *Gcc* across a wide range of sites and ecosystems from North America**, including
(a) a deciduous broadleaf forest at *queens*, (b) an evergreen broadleaf forest at *laupahoehoe*, (c) an evergreen needleleaf forest at
*austincary*, (d) a grassland *cperuvb*, (e) an agricultural site *mead1*, and (f) a shrubland site *luckyhills*.


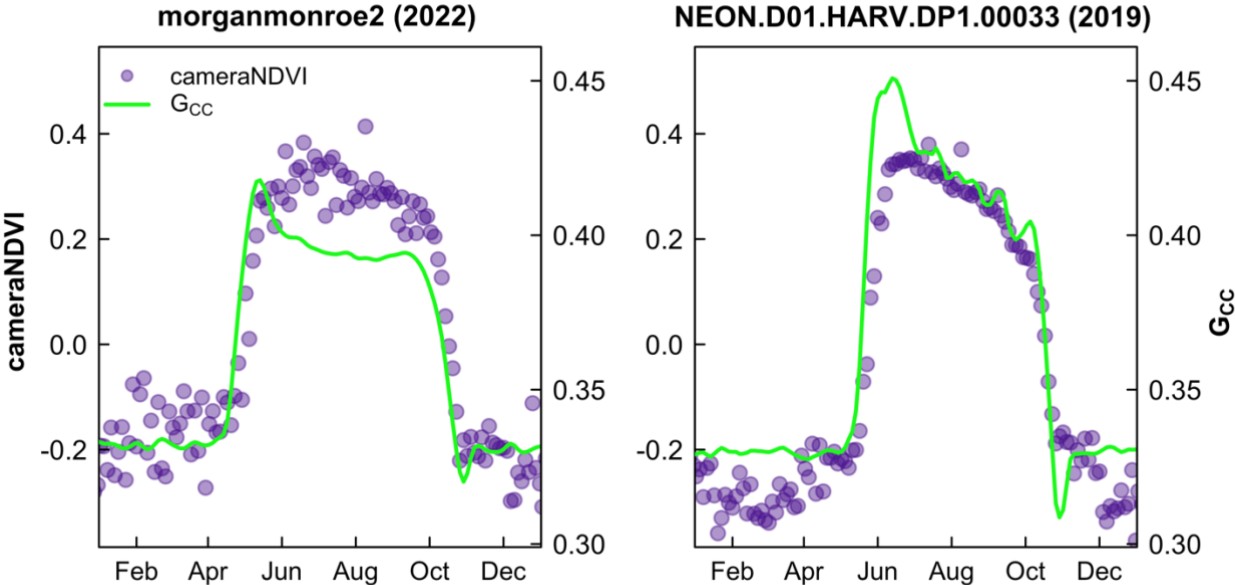

**Figure 4. Comparison of *cameraNDVI* and smoothed *G_CC* time series at deciduous broadleaf sites.** Note the early growing season greenness "spike" in *G_CC* that is absent from *cameraNDVI*. Scales on y-axis are equal for both sites.




While there was general agreement in the overall seasonality between *cameraNDVI* and $G_{CC}$, we found that in most cases, $G_{CC}$ provided clearer seasonal patterns and time series. For example, $G_{CC}$ provided much more distinct greenness signals in evergreen needleleaf forests relative to *cameraNDVI* (Fig. 3c). By comparison, an evergreen broadleaf site displayed similar levels of noise for both $G_{CC}$ and *cameraNDVI* (Fig. 3b). To summarize across all sites, we used a signal-to-noise ratio (SNR) analysis (Fig. 5), where we uncovered consistent evidence that $G_{CC}$ provides clearer seasonal patterns relative to *cameraNDVI*. Approximately 20% of all sites had *cameraNDVI* SNR estimates that were greater than SNR of $G_{CC}$; in other words, in almost 80% of cases, $G_{CC}$ provides a less noisy greenness metric for tracking phenology relative to *cameraNDVI*.




**Figure 5. Ratio of signal-to-noise ratio (SNR) of *cameraNDVI* to SNR of $G_{CC}$.** Top most panel shows cumulative distribution function of the ratio, where values < 1 indicate SNR for *cameraNDVI* is greater than SNR for $G_{CC}$, meaning less high-frequency variability in the *cameraNDVI* data and hence a less noisy *cameraNDVI* time series compared to $G_{cc}$. By comparison, values > 1 occur when $G_{CC}$ has less high frequency variability than *cameraNDVI*. From this analysis, SNR of *cameraNDVI* was higher than that of $G_{CC}$ for approximately 21% of site-years, whereas SNR of $G_{CC}$ was higher than *camera*NDVI for the remaining ~79% site-years. The bottom two panels show example time series for a site where $SNR_{DIFF} < 1$ (*tsubiology*) (deciduous broadleaf, DB) and $SNR_{DIFF} > 1$ (NEON.D10.CPER.DP1.00033) (grassland, GR). Note that in both cases, the time sereies are highly coherent between *cameraNDVI* and $G_{CC}$.





To bolster our evaluation of *cameraNDVI*, we further compared it to other reflectance-based measurements of canopy

greenness. Specifically, we compared *cameraNDVI* to *broadbandNDVI* derived from flux-tower data at NEON sites, which

revealed that there is generally a strong seasonal correspondence between the two datasets (Figs. 6, S2-S6). Overall,

*cameraNDVI* appeared to be less noisy than *broadbandNDVI*, and the clear correlation between these two datasets indicates

that *cameraNDVI* can provide a reliable—and perhaps better—greenness metric that is comparable to other estimates of

*NDVI*. In particular, *broadbandNDVI* exhibited some large outliers due to snowfall events. At times, *broadbandNDVI* was

highly variable from one day to the next, which is unlikely to be related to changes in canopy structure (Fig. 6). For example,

at some sites there appears to be a two-stage increase in early season *broadbandNDVI* (Fig 6a,h); in Fig 6a at Harvard Forest

– a deciduous broadleaf site – there is an early shift in *broadbandNDVI* likely due to initial snowmelt prior to leaf out in the

spring. Another example of noisier winter-time periods for *broadbandNDVI* can be found at Bartlett (a deciduous forest site

in New Hampshire, Fig. S1) and in tundra ecosystems of Alaska, such as Toolik (Fig. 6h) and Barrow (Fig. S2). By

comparison, *cameraNDVI* appears to be less sensitive to snow-covered time periods. We further compared *cameraNDVI* to

*broadbandNDVI* through a Signal-to-Noise Ratio (*SNR*) analysis at all terrestrial NEON sites, which was practically

identical to our *SNR* analysis between $G_{CC}$ and *cameraNDVI* (see Section 2.3). Through this *SNR* analysis, we found that

*cameraNDVI* was consistently less noisy at most of the NEON sites investigated (Fig. S7), further supporting our visual

evaluation that *cameraNDVI* is less prone to exhibiting extreme outliers and sensitivity to snow cover in colder climates than

*broadbandNDVI*.



**Fig. 6. Comparing cameraNDVI to *broadbandNDVI*.** NDVI estimates are calculated at co-located NEON towers.
Note that for all sites, the seasonality tends to be better defined in the *cameraNDVI* data compared to the
*broadbandNDVI* data.



## 4    Discussion

In this descriptor for the public data release of PhenoCam V3.0, we present significant updates to PhenoCam V1.0 and V2.0, published in 2018 and 2019, respectively (Seyednasrollah et al., 2019; Richardson et al., 2018b). In addition to more than doubling the total number of site years (Table 1, Fig. 1), we also significantly increased data availability in previously under-represented plant functional types, such as in forest understory ecosytems, evergreen broadleaf forests, grasslands, and wetlands (Sect. 3.1). Furthermore, we provide a new PhenoCam data variable: *cameraNDVI*, a measure of vegetation greenness that is conceptually similar to satellite and flux-tower based estimates of *NDVI* (Eqs. 1 and 3). To help guide users in applying *cameraNDVI* for scientific or educational purposes, we present the following discussion points on both the strengths and weaknesses of this new data variable compared to $G_{CC}$.

Through extensive tests directly comparing PhenoCam $G_{CC}$ and *cameraNDVI*, we ultimately found evidence that $G_{CC}$ provides a clearer and less noisy phenological signal of greenness compared to *cameraNDVI* at most sites (Fig. 5). In general, this more-variable signal in *cameraNDVI* can be attributed to a increased variance and a higher likelihood outliers occurring, a consequence of the following factors. First, large outliers can occur in *cameraNDVI* estimates; in particular, *cameraNDVI* < -0.5 seem to be associated with a stuck or cut infrared (*IR*) filter within the camera. Second, changes in lighting conditions during the calculation of *cameraNDVI* can cause a noisier signal. Specifically, since *cameraNDVI* is calculated from two images that are taken approximately one minute apart (one with IR filter and one without), *cameraNDVI* is subject to changes in lighting conditions during this 1-minute period (e.g., shifting cloud cover affecting incoming solar radiation), ultimately generating a noisier phenology signal relative to $G_{CC}$. Since, $G_{CC}$ is calculated from a single image, it is not sensitive to such changes. Finally, large outliers in *cameraNDVI* are also due to a higher sensitivity to snow than $G_{CC}$, leading to noisier data during the winter season, particularly in high-latitude ecosystems (e.g., DEJU and TOOL in Alaska, Fig. 6d,h). However, while $G_{CC}$ provides a less-noisy signal in general, this result is not ubiquitous across all condtions or vegetation types. For example, performance between *cameraNDVI* and $G_{CC}$ metrics appeared comparable at evergreen broadleaf (EB) sites (Figs. 3b, S1), and using our *SNR* analysis, we found that at ~55% of all EB sites *cameraNDVI* provided a cleaner signal than $G_{CC}$. To investigate specific outliers that may be due to snowcover in *cameraNDVI*, we suggest users visually inspect the image archive for the site in question, which can be browsed by year, month, or day. Imagery for each site is updated daily, and "site pages" can be accessed from the "gallery page" (<https://phenocam.nau.edu/webcam/gallery/>; for more information, see the tutorial on how to access PhenoCam data and imagery, available at <https://phenocam.nau.edu/education/PhenoCam_Access_Guide.pdf>). Furthermore, users can also access the archived imagery, from the ORNL DAAC in Ballou et al. (2025).

While *cameraNDVI*  exhibits several apparent weaknesses as a measure of phenology relative to $G_{CC}$ (i.e., noisier signal, higher sensitivity to snowcover), there are some key advantages that *cameraNDVI* may offer. First, *cameraNDVI* is more representative of seasonal LAI in deciduous broadleaf forest sites than $G_{CC}$. In particular, there is no distinct spring







"peak" in *cameraNDVI* (Fig. 4) (Keenan et al., 2014). Second, senescence derived from *cameraNDVI* is also delayed relative to $G_{CC}$ at the end of the growing season, such as in deciduous forests, likely representing changes in LAI rather than leaf color (Filippa et al., 2018). Similarly, at grassland sites, we found that the seasonal patterns of *cameraNDVI* are quite similar to $G_{CC}$, except *cameraNDVI* appears to decline more slowly in senescing grasslands because LAI remains high even if foliage is no longer green (e.g., Figs. 3d, 5). Finally, while *cameraNDVI* is not calculated directly from reflectance values – and therefore the absolute magnitude is not directly comparable to other *NDVI* measurements – *cameraNDVI* appears to give a cleaner phenology signal relative to flux-tower derived *broadbandNDVI* (Figs. 6, S2-S6). Overall, we encourage data users to view *cameraNDVI* as complementary to, but not a substitute for, $G_{CC}$; each index can provide unique information about different aspects of canopy development and changes in structure.



## 5 Data availability

Data are free and publicly available for download from the Oak Ridge National Lab Distributed Active Archive Center
(ORNL DAAC; <https://daac.ornl.gov>):

1. Digital Camera Imagery from the PhenoCam Network, 2000-2023:
   https://doi.org/10.3334/ORNLDAAC/2364 (Ballou et al., 2025)

2. Vegetation Phenology from Digital Camera Imagery, 2000-2023:
   https://doi.org/10.3334/ORNLDAAC/2389 (Zimmerman et al., 2025)

*For reviewers only, please access and download data using the anonymous login provided in the assets for review tab.*



# 6    Conclusions

Here, we present an updated version of the PhenoCam public data release (Version 3.0). PhenoCam V3.0 significantly expands the total number of site-years from 1783 in V2.0 to 4805.5 in V3.0. As with past releases, the imagery and time
series data have been quality-checked and controlled by a team of PhenoCam experts and data mangers, and all data and underlying imagery are freely and openly available. This version includes substantial updates to previously under-represented plant functional types, including evergreen broadleaf forests, grassland and agricultural sites, and understory vegetation (Table 1). In adidition to this expansion in available phenology data, we also include updates to the published Data Records. Specifically, we now include *cameraNDVI*, a metric of phenology that is based on infrared camera imagery
and provides a more direct comparison to other reflectance-based measures of *NDVI*, such as from satellites and or flux towers, relative to $G_{CC}$. Additionally, the new *simplified* time series and transition date data products included in the published PhenoCam Data Records should aid many users in both educational and basic research applications.



## 7    Information about the Supplement

Supplemental figures and tables that accompany this data descriptor are provided in the supplement section S1.





## 8   Author contributions

ADR initiated the PhenoCam network, obtained funding to develop and support the network, designed the observational protocol, and proposed the format of the standardized data sets. ADR and MAF oversaw project development. AMY led the efforts to draft this Data Descriptor with contributions from ADR. TM, KH, KLB, and MF contributed to systems design, development of processing routines, and management and maintenance of cyberinfrastructure. CC, KB, MJ, AKP, CS, ZV, OZ, BS and MM  contributed to dataset development, processing and curation, and evaluation and application. DMB, CRF, and MDS contributed to network development and cross-network collaboration. All authors reviewed and approved the content of this Data Descriptor.



## 9    Competing interests

The authors declare that they have no conflict of interest.





## 10 Acknowledgements

We thank our many collaborators, including site PIs and technicians, for their efforts in support of PhenoCam. Development and maintenance of the PhenoCam Network archive, has been supported by the National Science Foundation, the Long-Term Agroecosystem Research (LTAR) network which is supported by the United States Department of Agriculture (USDA), the U.S. Department of Energy, the U.S. Geological Survey, the Northeastern States Research Cooperative, and the USA National Phenology Network.

The views and conclusions contained in this document are those of the authors and should not be interpreted as representing the opinions or policies of the U.S. Geological Survey or the National Science Foundation. Mention of trade names or commercial products does not constitute their endorsement by the U.S. Geological Survey or the National Science Foundation.

The National Ecological Observatory Network is a program sponsored by the National Science Foundation and operated under cooperative agreement by Battelle. This material is based in part upon work supported by the National Science Foundation through the NEON Program.

Funding for the AmeriFlux data portal was provided by the U.S. Department of Energy Office of Science.





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
