# Peer review of "Tracking vegetation phenology across diverse biomes using Version 3.0 of the PhenoCam Dataset"

_Earth System Science Data, 2025_

## Author Comment (AC1)

**Response to reviewers and community for essd-2025-120**

We thank both reviewers and the community member who provided thoughtful and helpful comments on the manuscript "Tracking vegetation phenology across diverse biomes using Version 3.0 of the PhenoCam Dataset" by Young and coauthors. By addressing these comments, we believe the manuscript has been significantly improved and we are excited to share our responses and our manuscript revisions.

Please note, our responses are provided in the attached PDF in the following format:

- At the top level, the comments section from each reviewer (i.e., RC1 and RC2) and community member (CC1) starts at the top of new page which are indicated by **Bold Underline**.
- *Individual comments from RC1, RC2, and CC1 are italicized.*
- Author replies in response to each comment are in dark blue and numbered in the format **AC1**, **AC2**, etc.
- Revised or added text in the manuscript is in orange.
- Line numbers refer to the revised manuscript, which has been uploaded separately.
- Revisions in the manuscript document are in dark blue.

Sincerely,
Adam Young

**Community Comment 1 (CC1, Mukund Palat Rao)**

*This is a nice summary paper of the PhenoCam V3.0 data set. Thanks Young et al. and all the PhenoCam contributors for your hard work in providing and maintaining this invaluable community resource.*

*A comment that I think might help: many users of regular reflectance based NDVI are used to a certain range of values, e.g. 0.2-0.5 sparse vegetation, >~0.5 dense vegetation., and negative for non-vegetated surfaces. So I think it might be useful for explicity dicuss why cameraNDVI values are negative/can be negative. Even better if you could provide some kind of scale or rubric for where different vegetation types would be on the PhenoCam cameraNDVI range. Looking at Fig. 3, cameraNDVI almost never exceeds 0.2 at its peak (dense vegetation?), around -0.15 to 0 for modrate vegetation. At least, clearly mention that negative or zero cameraNDVI does not mean 'no vegetation'. Live vegatation at SH has NDVI values around -0.4.*

*One place where this could be mentioned: Lines 498-500: Finally, while cameraNDVI is not calculated directly from reflectance values – and therefore the absolute magnitude is not directly comparable to other NDVI measurements – cameraNDVI appears to give a cleaner phenology signal relative to flux-tower derived broadbandNDVI (Figs. 6, S2-S6).*

**AC1:** We sincerely thank the reader for their thoughtful comment. Both reviewers also noted negative cameraNDVI values as a point of concern, so this is a key issue to address in our manuscript revision. We have added the following paragraph on L556-565 to help both help explain negative cameraNDVI and note that it does not mean "no vegetation." It reads as follows:

Prior to discussing comparisons between GCC and cameraNDVI, we note that cameraNDVI are often negative (i.e., < 0), even during periods with green vegetation in the field-of-view (e.g., Fig. 4f). This is an important distinction when compared to the more common physical interpretations of NDVI derived from satellite remote sensing (Eq. 1). Negative values most likely emerge from the fact that cameraNDVI is calculated from exposure-adjusted pixel intensities, rather than true measures of reflectance. Although intensity has been shown to scale with reflectance for both the R and NIR channels (Petach et al., 2014), the relative magnitude of R vs NIR pixel intensity does not necessarily correspond to the relative magnitude of R vs NIR reflectance. Consequently, while seasonality of cameraNDVI may correctly depict seasonal vegetation dynamics, the absolute magnitude of cameraNDVI may be quite different from standard NDVI products from satellite platforms. To facilitate comparisons across sites, one potential solution is to re-scale cameraNDVI to match the range of satellite NDVI (e.g., MODIS), as suggested by Filippa et al. (2018).

*This manuscript by Young et al. presents Version 3.0 of the PhenoCam dataset, a comprehensive, multi-year, and multi-biome collection of near-surface remote sensing data for vegetation phenology monitoring. The authors provide a robust description of dataset structure, methodology, and applications, along with valuable comparisons between GCC, cameraNDVI (derived using PhenoCams), and broadbandNDVI (using typical net radiation and PAR sensors). This paper builds on and extends the PhenoCam 2.0 dataset, which is already a tremendous resource for scientists across a range of scientific communities. A new aspect of PhenoCam 3.0 is that it enables the utility of cameraNDVI data and a simplified data structure. Importantly, this work addresses not only a clear need for high-frequency, ground-based phenology observations that complement satellite datasets and support model validation but provides a well-documented community resource.*

*Major Comments:*

*One thing that was a bit unclear to me was regarding the addition of Data Record 7 (simplified products). It may help to clarify in the main text whether these simplified files include uncertainty estimates or metadata, and if not, whether that may impact scientific use.*

**AC2:** We thank the reviewer for this comment and feedback! In general, the main motivation for producing the simplified files is to provide users with a streamlined and direct way to retrieve the most used variables that also have the broadest inter-discipline accessibility: gcc_mean and transition_dates. Based on how past users seem to access and use the data, we believe the uncertainty estimates will be more geared towards "power users," or those with a higher expertise in data analysis and statistics. The simplified format of the files in data_record_7 also removes all the additional header and metadata, making it much simpler to import and almost immediately start using in any software program designed for data analytics.

To help address your comment, we have added an additional note on L153-154 highlighting that we DO NOT include any associated metadata or uncertainty estimates in the simplified file. It reads as follows: For users who wish to access additional information, such as metadata or uncertainty estimates, these can be found in Data Records 3, 4 and 5.

*Otherwise, I only have minor comments. Overall, the paper is well-written, methodologically sound, and makes a significant contribution that is likely to be widely used. It is well-suited for publication in Earth System Science Data with minor revisions.*

*Minor Comments:*

- *Around line 100 – perhaps mention the issue of clouds as well. Satellites can't see through them, but ground based data can fill these gaps.*

  **AC3:** We thank the reviewer for this comment and we have now added additional text on ground-based data filling in potential gaps caused by cloud cover on L102-104. It reads as follows: Additionally, extensive cloud cover – particularly for multiple days or weeks – obscures and reduces the ability of satellites to detect changes in vegetation, indicating the

ability of near-surface remote sensing methods to provide time series with fewer gaps (Tran et al., 2022).

- Line 145- The description of the simplified GCC products should briefly clarify that while uncertainty is omitted, users may refer to Data Record 5 for uncertainty quantification if needed.

  **AC4:** Thank you, this is now addressed on L149-150 and a copy of this new text can be found in **AC2**.

- *Line 149 – no need to capitalize phenology here.*

  **AC5:** Thank you for pointing this out. We have now made this change.

- *Line 194: Do end-users have access to the 'exposure time' data? Or how is this extracted? Is it a pre-set exposure, or does it automatically adjust based on irradiance conditions? Potentially some more information would be useful. \*\*\*I now see from Box 2 it's provided in the metadata. Do all sites have all of this metadata associated with them?*

  **AC6:** Yes, as the reviewer noted at the end of their comment, this image metadata is provided in the ndvi_roistats file for both RGB and NIR images, and is pulled from the internal camera settings at the time each image is recorded (Box 2). The ndvi roistats file is part of Data Record 6 and can be found with every site that has an IR-enabled camera and where cameraNDVI is calculated.

- *Around line 220: Perhaps mention the FOV is going to be a bit different between broadbandNDVI and cameraNDVI. How so?*

  **AC7:** We have now added text on L240-244 clarifying that the FOV differences between cameraNDVI and broadbandNDVI. It reads as follows:

  …, and these issues are minimized for broadbandNDVI measurements obtained from the same tower where PhenoCams are mounted. It should be noted that the comparison between cameraNDVI and broadbandNDVI is not perfectly aligned due to field-of-view (FOV) differences: PhenoCams have an oblique FOV of the canopy, while both photosynthetically active radiation (PAR) quantum sensors and shortwave pyranometers have a hemispherical FOV and a cosine response.

- *Figure 1: Great figure. Might be nice to also have a smaller dot in the center of each larger transparent circle so we can hone in on exactly where these data come from. Probably not needed but an idea.*

  **AC8:** Thank you! We appreciate the complement and feedback on adding the exact site locations to each circle. However, we believe the current figure should remain unchanged for two reasons:
  1. We provide this exact same figure in all three data descriptors (V1.0, V2.0, and V3.0); therefore, we would prefer to keep as is so it can be more directly

compared with past versions and make it easier to identify the growth of the network.

2. The circles are scaled and aggregated per 4x4 degrees latitude and longitude, so plotting each site would not necessarily be in the center of each circle and potentially risk making the figure "busier" and less easy to interpret.

- *Figure 3: The camera NDVI values are quite low. Especially in SH, where they are all negative. Should this be discussed? Either way, these results are impressive, and makes you wonder, do we need NDVI at all? The GCC data seem to be more dynamic and 'sharper'. \*\*ah yes, just as you mention around line 425.*

  **AC9:** This is a really important point that was brough up by a community comment and the other reviewer. To address and better explain negative cameraNDVI values, we have added a paragraph on L556-565. Please refer to **AC1** in this document for a copy of this revised text.

- *Figure 5: Very nice, the SNR analysis is great here and gives ideas for future researchers.*

  **AC10:** This is a very nice piece of feedback to receive, thank you! We agree and believe it is an efficient method for comparing time series from two different sensors across a large network.

- *Line 485 – yes, always visually inspect the data!*

  **AC11:** Thank you, yes, we strongly agree! We have found the PhenoCams to be quite useful for visually inspecting and interpreting other data streams as well (e.g., sensors from flux towers), and this information likely represents an underutilized strength of the dataset. We also highlight this a second time in our new paragraph at the end of the discussion (see response to your next comment, **AC12**).

- *Note that there is a growing number of near-surface remote sensing platforms (see recent Tansley Review by Pierrat et al., 2025 https://nph.onlinelibrary.wiley.com/doi/epdf/10.1111/nph.20405 ) and it might be nice in the discussion to discuss briefly how PhenoCam can serve to compliment these data sources, or how it can be used as an example for how future networks might consider standardizing and producing user-friendly data products.*

  **AC12:** reviewer 2 had a similar comment regarding strengths and weaknesses of the PhenoCam data products. We have provided a paragraph at the end of the Discussion that addresses both (L598-617), and reads as follows:

  Through standardized data collection and processing protocols, as well as the continually growing size of the network, PhenoCam data products offer a powerful tool to study vegetation phenology in almost any terrestrial biome (Richardson et al., 2013; Richardson, 2023). As with any environmental data product, there are key strengths and caveats that users must consider. First and foremost, PhenoCam GCC captures changes in leaf pigmentation and canopy color, which frequently aligns very closely with

photosynthetic phenology (Bowling et al., 2018; Keenan et al., 2014), and can also provide clear and consistent estimates of phenological transitions (Richardson et al., 2019; Dunn et al., 2022). However, as discussed in the previous paragraph, GCC is less capable of capturing changes in canopy structure and LAI; by comparison, the new cameraNDVI product appears to offer a better measure of canopy structure. Furthermore, GCC is relative at each site; individual sites are influenced by both the color of foliage and the amount of background visible through the canopy, leading to variability when comparing the magnitude of GCC values between sites. Finally, one of the most important strengths of PhenoCam is the standardized collection and data processing of repeat imagery from across the observatory. This standardization is critical for multiple reasons: (1) it produces a consistent visual record of site and environmental conditions, (2) it allows the monitoring of fine-scale or short-term changes in vegetation (e.g., Knox et al., 2017; Hufkens et al., 2012), (3) provides a framework for conducting regional-continental scale syntheses and evaluation of satellite remote sensing products (e.g., Young et al., 2022; Moon et al., 2019; Bolton et al., 2020), and (4) the scale and footprint of PhenoCam data are well aligned with other near-surface ecological datasets, such as eddy covariance towers (e.g., Oishi et al., 2018; Desai et al., 2022; Liu et al., 2025), thermal cameras (e.g., Javadian et al., 2024), SIF (e.g., Zhang et al., 2023; Magney et al., 2019), and LiDAR (e.g., Musinsky et al., 2022). To date, by leveraging the strengths of standardized processing routines and community engagement, PhenoCam data products have been cited and used in approximately 500 publications over the last 17 years (Richardson and Javadian, 2025).

- *This is a great contribution, thank you for all your efforts to keep PhenoCam alive, it has had enormous benefits to the scientific community.*

  **AC13:** We sincerely thank the reviewer for their kind and thoughtful comments, we truly appreciated their review!

*Young et al. provide a substantial update to the Phenocam Dataset, with V3.0 containing expanded site/biome and temporal coverage, and the addition of a new variable, cameraNDVI and simplied summary daily files that will be useful for many/most scientific applications as well as for for education and public outreach. They present a summary of the expansion to the network, explain the data records for the new variables, and then conduct an analysis of the new cameraNDVI variable. They find it is more sensitive to changes in leaf area, while Gcc is more sensitive to pigment changes. They also explore the variance in the cameraNDVI compared to both the Gcc and broadbandNDVI calculated from measurements of incident and upwelling solar radiation at Ameriflux sites. They find the cameraNDVI gives a clearer phenology signal compared to broadbandNDVI and note that it should be viewed as complementary to Gcc given the different information in the signal.*

Major Comments:

*This is a very informative overview of the new dataset for current and new users of PhenoCam data. The paper was well written, the scope and objective of the paper were clearly defined. The new data and methods clearly described. The results are nicely presented. In the discussion the authors focus on the strengths and weaknesses of the cameraNDVI, which is helpful for researchers who want to use this new variable.*
*I have worked on this review with a student that is new to my lab – and to this topic – and they found the review paper was very clearly written and easy to understand. They were the one to work with the data for the start of their research project. They said working with the data was easy. However, they did note that accessing V3 data is less efficient due to the lack of a queryable structure. Without predefined indexing mechanisms, all 18,102 files are stored in a single directory, making it an intensive process to parse and extract relevant information. As a result, any program built to work with V3 will face increased time and space complexity, especially when scanning, filtering, or preprocessing the data for specific sites or time ranges.*

*The student accessed the V2 data through the PhenoCam Network's official download page: https://phenocam.nau.edu/webcam/network/download/, which links to the API documentation. https://phenocam.nau.edu/api/ and https://phenocam.nau.edu/api/docs/. They said the PhenoCam API provides structured, programmatic access to V2 data, organized into clearly defined endpoints, including site metadata, image data, and processed summaries like daily counts and midday images. This made V2 far easier to much easier to query, navigate, and integrate into workflows compared to the more bulk-style V3 data release. I am guessing that v3 will eventually be available through the PhenoCam API?*

**AC14:** We thank both the reviewer and their graduate student who tested the accessibility and usability of PhenoCam V3.0 on both ORNL and the PhenoCam webpage/API.

We agree with the reviewer. The way in which the data are made available through the ORNL portal is not ideal. It is also not how we would have chosen for the data to be distributed, but ORNL did not want the data for each site or ROI to be in zip files. We describe our solution to this below by offering several methods to make it easier for users to explore and access the V3.0 through links on

the PhenoCam webpage, while also noting that archiving at ORNL ensures long-term data availability.

We also recognize the enhanced accessibility offered by the API on the PhenoCam server. We would also like to note that the PhenoCam team has evolved and changed over the years, and since the release of V2.0 there have been several key team members retire. As a result, we are not currently aware of a way to directly access the curated V2.0 datasets through the API (but if your student has found a way to do so, we would appreciate knowing how this was done!). We think it is more likely, however, that they were accessing the provisional data sets. We have clarified this in the manuscript, and we present new text at the end of this response.

Overall, to help address the reviewers well thought out concerns regarding ease of access to the data, we have made two major changes:

1. We have added several features on the PhenoCam webpage to make it easier to directly download and explore the V3.0 data for a given site or all sites. Specifically, we now include:
    a. Updated database tags to indicate at the top of each site page whether it has been included in a particular release which then links to the ORNL archive (See below new Fig. 1c, red box #8).
    b. The entire V3.0 dataset is now available to download as a single zip file via PhenoCam Explorer (See below new Fig. 1a, red box #1) or a dropdown menu from the PhenoCam webpage (See below new Fig. 1e, red box #11). This zip file contains the data for each unique ROI which are in turn packaged in their own zip files, with each containing folders for the seven data records described in the manuscript and past descriptors.
    c. Individual zip files for specific sites can now also be accessed from PhenoCam explorer or the PhenoCam webpage (new Fig. 1b box #3, new Fig. 1e box #12). To download individual zip files using PhenoCam Explorer requires selecting an ROI and then clicking download (there is no direct URL link to the data files). However, on the main PhenoCam web page, we provide a downloadable CSV containing a table of ROIs and the URL to obtain the associated data file (https://phenocam.nau.edu/data/releases/v3/by-roi/PhenoCam_V3_roilist.csv), making it straightforward to script a large number of downloads in programming languages such as R or Python.
2. To accompany these new ways to access V3.0, we have added a new section (including figure) to the methods that describes in detail the multiple routes a user can go through to access V3.0 (and PhenoCam data in general). This new section is found on L362-407 and reads as follows:

**2.5 Accessing PhenoCam V3.0**

The PhenoCam V3.0 data release can be accessed three different ways:

1. The Oak Ridge National Laboratory Distributed Active Archive Center (ORNL DAAC), which is free to use and access (registration for an EarthData login is required). This archive also includes a helpful User Guide to better understand the dataset structure and organization. Please see the Data Availability Statement in Sect 5.

2. The PhenoCam Explorer webpage (Fig. 1a,b). This webpage (<https://phenocam.nau.edu/phenocam_explorer/>) is free to use, and offers several tools to query, search, and visualize the PhenoCam V3.0 data products for each site. Users can access and evaluate previous versions of PhenoCam data releases (V1.0 and V2.0) through this portal as well. This page includes a button allowing users to download the entire V3.0 dataset as a single zip file (Fig. 1a). This zip file contains the data for every ROI in V3.0 packaged in their own individual zip files, each containing directories for each of the seven data records described in this paper, Richardson et al. (2018b), and Seyednasrollah et al. (2019) (Sect 2.4). The PhenoCam Explorer webpage also offers options to download versioned zip files for individual ROIs (Fig. 1b).

3. The PhenoCam Gallery (<https://phenocam.nau.edu/webcam/>, e.g., Fig. 1c,d,e). At the top of the PhenoCam webpage, there are several persistent dropdown menus that offer links to download the data or visit the Explorer webpage. On each individual site page (e.g., Fig. 1c,d), users also have access to links indicating if a site is part of a data release, and each link points the user to the ORNL data archive. The "Download PhenoCam V3 Dataset" link noted at the top of Fig. 1d takes the users to Fig. 1e, where they also have the option to download the entire V3.0 dataset as a single zip file (~6.5 GB), or to download zip files for individual V3.0 ROIs. This page (Fig. 1e) also offers the option to download a list of all ROIs and associated zip file URLs to aid in programmatic access (e.g., via R or Python) to the versioned data. To download data via this page users must first register with PhenoCam (which is also free). Finally, under the URL for each ROI (Fig. 1d), users can access additional information (e.g., visualization of ROI mask or time series of GCC) and download the provisional data.

We encourage users to explore some or all these pathways for accessing V3.0 to find the option that will best suit their own research or education requirements. Finally, it is critical to note the difference between versioned and provisional data sets: versioned data (i.e., V1.0, V2.0, and V3.0) are prepared for long-term archive at ORNL, have undergone extensive QA/QC, and are static (i.e., they will not be changed in the future), ultimately making them ideal for conducting reproducible science. By comparison, provisional datasets accessed through the PhenoCam gallery and API contain results from the most recent data acquisition and are updated daily but have not undergone the same quality checks and review after the end date of V3.0 (i.e., 2023-12-31).

[Figure]

**Figure 1. Various ways to access PhenoCam V3.0 data.** (a) Using PhenoCam Explorer (<https://phenocam.nau.edu/phenocam_explorer/>), users can explore the spatial distribution of available PhenoCams in V3.0. There is also the option to download the entire V3.0 dataset as a zip file (red box #1). Red box #2 indicates the "Plot and Download Data" tab, which takes users to (b) and allows for broader query

options for specific sites or vegetation types, as well as exploring visualizations of time series, transition dates, and relationships with other variables (NDVI, EVI). This page also offers a download button on the bottom for each specific ROI (red box #3), which will provide a zip file of the V3.0 data for that specific ROI. Red box #4 takes users to (c) the landing page for a given site in the PhenoCam gallery. The PhenoCam gallery webpage (<https://phenocam.nau.edu/webcam/>) has a persistent header of drop-down menus, providing links to visit the explorer page (Fig. 1c, red box #5), the Application Programming Interface (API, Fig. 1c, red box #6), or to download V3.0 data (Fig. 1d, red box #7). For each individual site page in the PhenoCam gallery, we provide metadata at the top indicating which versioned data releases the site is included in, pointing users to the ORNL archive (red box #8). The ROI link(s) for each site (red box #9) take users to (d), which provides additional information and a link to download provisional data (red box #10). The "Download PhenoCam V3 Data" link under the drop down menu (red box #7) will take users to the PhenoCam V3 Release ROIs page (e) where there are additional options to download the entire V3.0 archive as a single zip file (red box #11) or download zip files for individual ROIs (e.g., red box #12). Finally, in Fig. 1e (red box #13), users have the option to download a CSV table that contains V3.0 versioned zip file URLs for each ROI to aid in programmatic downloads of the V3.0 dataset.

*Beyond that question, I have only minor comments that are provided below and a question about negative camerNDVI at the shrub and grass sites provided as examples. Otherwise, I think this is really excellent dataset of ground based phenology that will be useful for a wide range of research/scientific applications and for education and outreach. I look forward to seeing this paper published.*

**AC15:** Thank you for the kind words and helpful comments on our manuscript!

*Minor comments*

*Introduction*

- *Lines 117-119: could also consider Yan et al. (2019)*

    **AC16:** Thank you for this suggestion, we have made this change.

*Methods and Materials*

*Section 2.4:*

- *I appreciate data records 1 to 5 have been described previously and the authors don't want to repeat here. But as I'm reading data record 6 I'm wondering if the RGB ROI statistics are the same as in data record 3? And if so, why repeat? I guess for transparency for the camera NDVI calculation?*

    **AC17:** Yes, exactly. In the NDVI stats file (Box 2) we provide all metadata and statistics to calculate the cameraNDVI value.

- *I think it is the RGB stats are same from looking at the user guide on the V3 ORNL DAAC website, which as an FYI I found very useful for explaining all this, so perhaps refer readers to that document as well?*

**AC18:** Thank you for this suggestion. We have now mentioned the ORNL User Guide on in the new Data Access Section 2.5. We also mention an additional tutorial available on the PhenoCam webpage on L583-584 in the Discussion.

- *It is very helpful to see an example of how a file will look.*

   **AC19:** We thank the reviewer for this comment and agree that being able to visualize 1- or 3-day file would be nice (and this same topic was briefly discussed among co-authors prior to submission). We made the decision to exclude it given large number of columns; it is just very difficult to provide a usable example within the paper. We do provide the header and three sample lines for the ndvi files in Boxes 2 and 3, and headers and sample lines are provided for the gcc and transition date files in the V1.0 data descriptor (Richardson et al., 2018b).

- *It is very helpful to have so many stats already calculated. Are there uncertainties provided for cameraNDVI?*

   **AC20:** Uncertainty estimates are not provided for cameraNDVI. This is because uncertainty is derived from the spline interpolation smoothing algorithm implemented in the *phenocamr* package, which is not currently applied to *cameraNDVI* given the generally noisier signal (i.e., see SNR analysis in manuscript). However, we do provide standard deviation of cameraNDVI across all images that pass the initial QC filters (e.g., not too dark, sun > 5 deg above horizon) in the ndvi roistats file (Box 2).

- *Data record 7 is an excellent idea. I appreciate the idea is to keep things simple, but given the purpose can be for scientific applications (in addition to education or science outreach) why not also include the same two variables for cameraNDVI?*

   **AC21:** Thank you for this suggestion. We had a similar comment from reviewer 1 about including more variables in the simplified file. At this time, the simplified files are aiming to provide a very straightforward structure for easy access to the most frequently used PhenoCam derived data: transition dates and gcc_mean, and smooth_ gcc_mean. As cameraNDVI is a new product, we believe it practical to first see how the user community adopts to and uses it before including it in the simplified files.

*Section 3 (Results)*

- *In Table 1 I think it would be useful to have the number of sites as well as site-years?*

   **AC22:** Thank you and we agree. We have now added number of sites to Table 1. It has been modified and now appears as follow:

**Table 1. Vegetation type abbreviations for ROIs (region of interests), and the corresponding number of sites and site-years of data in the PhenoCam dataset described here (V3.0).** For comparative purposes, the number of sites and site-years of data in the previous dataset releases is also presented. The number of sites that contain an ROI for a given vegetation type are in parentheses, and a given site can contain ROIs for multiple vegetation types. MX and NV ROIs were

excluded in V2.0 but are currently available again in V3.0. There are 2.7 site years of Reference Panel (RF) ROIs in V3.0 as well, for a total of 4805.5 site years in the V3.0 data release.

| Abbreviation | Description | Site-years ($n_{sites}$) in V1.0 | Site-years ($n_{sites}$) in V2.0 | Site-years ($n_{sites}$) in V3.0 |
|---|---|---|---|---|
| AG | Agriculture | 50 (11) | 226 (84) | 703.5 (161) |
| DB | Deciduous Broadleaf | 392 (67) | 653 (112) | 1185.2 (171) |
| DN | Deciduous Needleleaf | 4 (1) | 45 (11) | 115.3 (13) |
| EB | Evergreen Broadleaf | 2 (1) | 28 (12) | 101.8 (22) |
| EN | Evergreen Needleleaf | 80 (18) | 264 (66) | 778.0 (122) |
| GR | Grassland | 121 (26) | 280 (70) | 912.4 (188) |
| MX | Mixed vegetation (generally EN/DN, DB/EN, or DB/EB) | 5 (1) | - | 13.7 (2) |
| NV | Non-vegetated | 14 (1) | - | 17.2 (3) |
| SH | Shrubs | 46 (13) | 141 (48) | 436.8 (86) |
| TN | Tundra (includes sedges, lichens, mosses, etc.) | 22 (7) | 68 (15) | 117.0 (20) |
| UN | Understory | - | 18 (10) | 219.2 (41) |
| WL | Wetland | 11 (4) | 58 (20) | 202.7 (39) |

- *I think it could be really useful to have an additional table after Table 1 that contains the number of sites and site-years for each of the Level I ecoregions of North America. Then if researchers are focused on one or two specific regions they will be able to see the increase in number of sites and site years for that?*

  **AC23**: Thank you for suggesting this. We have now added another table to complement Table 1 that has number of sites and site years for each Level I Ecoregion:

**Table 2. Number of sites and site years for each Level I Ecoregion in North America.** These Level I ecoregions correspond to the same ecoregions in Fig. 2 (Omernik and Griffith, 2014). Please note, not all site years/sites are included if they are located outside North America.

| Abbreviation | Description | Site-years ($n_{sites}$) in V1.0 | Site-years ($n_{sites}$) in V2.0 | Site-years ($n_{sites}$) in V3.0 |
|---|---|---|---|---|
| EF | Eastern Temperate Forests | 313.5 (40) | 617.3 (61) | 1382.7 (182) |
| GP | Great Plains | 36.0 (10) | 165.4 (27) | 492.4 (79) |
| MC | Mediterranean California | 63.2 (15) | 98.4 (15) | 199.5 (38) |
| ND | North American Deserts | 29.4 (11) | 66.2 (17) | 412.4 (107) |
| NF | Northern Forests | 153.0 (28) | 468.2 (44) | 1006.4 (86) |
| NW | Northwestern Forested Mountains | 87.1 (15) | 165.3 (30) | 375.4 (55) |
| SA | Southern Semiarid Highlands | 6.1 (4) | 14.0 (4) | 62.1 (6) |
| TG | Taiga | - | 3.6 (1) | 25.2 (6) |
| TN | Tundra | 26.1 (7) | 50.3 (10) | 75.7 (14) |
| TS | Temperate Sierras | - | 3.2 (3) | 126.7 (26) |
| WC | Marine West Coast Forest | 7.4 (2) | 18.2 (6) | 41.3 (10) |

- *Figure 3: Interesting time series comparisons. I'm surprised by negative cameriaNDVI for much of the time series for the grass and shrub site though as there clearly is green vegetation there? Having said that, the authors do mention later that the values are not comparable to NDVI calculated from reflectance. I've also read another reviewer comment that points out that even when NDVI is above zero it's not as high as we might expect with NDVI values we're used to seeing from satellites. Still, the negative NDVI is a little surprising. I guess I should go and read the Petach et al. (2014) and Filippa et al. (2018) papers to learn more. If I'm remembering correctly the Wingate et al. reference I mention in my next comment addresses the issue of using DNs. But I agree with the other reviewer that explainig this a bit more when Fig. 3 is presented in the results. (while also pointing to the earlier papers) might be helpful.*

  **AC24**: Thank you for bringing this up. Not including a more detailed explanation of negative NDVI values was clearly an oversight in our initial draft. We have included a paragraph (L556-565) in the Discussion that describes in more detail why negative cameraNDVI values may be more common. A copy of this new text can be found in **AC1** in this document.

- *Lines 404-410: Wingate et al. (2015) would be a good reference here. They used PROSAIL to show RGB signals/fractions across the European phenocam network were sensitive to chorolphyll and other pigments (and to some extent LAI), while NDVI is more sensitive to LAI (Section 3.2.1 of that paper).*

  **AC25:** We have now added this reference, thank you.

- *Figs S2 to S6 are referenced before S1. And I'm not sure the reference for Fig. S1 at Line 449 is correct? I think that should be S2?*

  **AC26**: Thank you for pointing out these mistakes. We have now corrected both.

- *Also line 449: Barrow reference should be Fig. S6.*

  **AC27**: Thank you for pointing out this mistake. We have now corrected it.

- *Figure 5 and S7: couldn't hurt to have SNR_Gcc / SNR_cameraNDVI (and equivalent for S7) in the x-axis label in parentheses.*

  **AC28**: We agree, this is a good suggestion and appreciate it. We have made this change. We have also added to now Fig. 6 by plotting where the SNR_diff values would fall along the probability plot in the top panel for both example sites (with labels included), as well as providing the actual SNR_diff values in the figure legend for those two sites.

- *Perhaps a correlation analysis across all sites could be added between the Gcc and cameraNDVI time series so we can see across the huge range of sites which have a strong correlation or not (actually same for the SNRdiff analysis) with table summarising per ecoregion/vegetation type, or a map with point size in proportion to the correlation or SNR diff so we can see which ecoregions/vegetation types tend to have a closer correspondance (more or less variance in one or the other variable)? This would complement the examples shown in Figs. 3 to 5? Do all evergreen needleleaf sites have a clearer phenology signal in Gcc as shown in Fig. 3c?*

  **AC29**: We thank the reviewer for this comment and helpful suggestion on how to visualize the relationship between Gcc and cameraNDVI. We have address this by:

  1. Calculating the correlation between Gcc and cameraNDVI for all sites where there is at least one year of overlap between the two variables.
  2. Summarized the distributions of these correlation results for each vegetation type and Level I ecoregion in a new Figure (Fig. 7) with sets of box plots. The new text and figure are presented below.

  **Methods text** (L219-222): Finally, we further explored the relationship between GCC and cameraNDVI by each individual plant function type (PFT, see Table 1) and Level I Ecoregion (see Fig. 2, Table 2) through boxplots that compare the distributions of both (1) linear correlations between GCC and cameraNDVI 1-day time series, and (2) SNR(GCC)/SNR(cameraNDVI).

  **Results text** (L504-508): When separating this analysis by individual PFTs and Level I Ecoregions, we found similar patterns where cameraNDVI was in general noisier than GCC (Fig. 7). There were a few notable exceptions; in 63% of all evergreen broadleaf (EB) sites, cameraNDVI had a less noisy signal relative to GCC (Fig. 7c). Shrublands (SH), grasslands (GR), and evergreen needleleaf (EN) forests displayed an opposing pattern compared to EB sites, with only 8%, 9%, and 11% of sites where cameraNDVI was less noisy than GCC, respectively.

[Figure]

**Figure 7. Summarizing relationships between GCC and cameraNDVI by vegetation type (i.e., PFT) and Level I Ecoregions.** In (a) and (b), the distributions represent the linear correlation between 1-day time series for GCC and cameraNDVI. In (c) and (d), distributions represent the signal-to-noise ratio (i.e., SNR Difference = SNRGCC / SNRcameraNDVI), where values > 1.0 indicate that GCC has a less noisy or smoother signal.

*Section 4 (Discussion)*

- *Line 475: The authors identify NDVI < -0.5 is due to IR filter issues, but it is unclear whether such values were filtered or flagged in the dataset. This should be more explicitly discussed in the Methods or Data Records sections.*

  **AC30:** We thank the user for bringing up this question. We do not do outlier detection or other filtering for cameraNDVI, and there has been only minimal curation of the data. The image-to-image and day-to-day variability in cameraNDVI, as well as across-site variability in cameraNDVI, has challenged our attempts to implement procedures similar to what we use for GCC, and it is for the same reason that we do not extract transition dates from the

cameraNDVI time series. We hope that by drawing more attention to the availability of cameraNDVI as a PhenoCam data product we will get community feedback on what would be most useful for data users, and then we would hope to implement these suggestions into our processing pipeline for a future V4.0 data release. We have added the following brief text on L203-205 in the Methods (Sect 2.2) to highlight that there is no outlier detection yet for cameraNDVI: Additionally, there is no outlier detection mechanism implemented for cameraNDVI, given challenges with the higher variance of this data product (see Sect 3.2). This remains an ongoing area of research and development that will be implemented when available.

- *Line 485: But there is also a snow flag, so is this just to have an additional verification? Does it mean the snow flag not reliable?*

  **AC31:** This is a good question regarding the snow flag. When we started preparing the V1.0 dataset in 2015, our plan was to use crowd-sourced classification of images (snow vs not snow) to develop a deep learning model that could automatically flag snowy images (see https://journals.plos.org/plosone/article?id=10.1371/journal.pone.0209649). Although the model was never applied as part of our routine processing, the snow flag column remained in the dataset.  As we have published documentation for V1.0, V2.0, and V3.0 datasets, we have tried to maintain overlap between publications - which meant that it was easier to leave the snow flag column in the "ROI stats" image file, than to delete the column and explain this change to the data format.

- *I was also wondering whether there are other QC flags (for clouds for example), but I guess this is all tied up in the Type I vs Type II vs Type III datasets?*

  **AC32:** Thank you for bringing this to our attention. There is no filtering for clouds. Our method of aggregating from tens of images per day in the ROI stats file to the 1- and 3- day summary files is designed to minimize the impact of adverse weather conditions (see Sonnentag et al 2012). We now briefly reference this on L263-269 and refer readers to V1.0 and V2.0 for specific details: The PhenoCam Dataset V3.0 contains seven separate Data Records for each site (Box 1). The structure for Data Records 1-5 is unchanged and described in detail in the data descriptors for V1.0 (Richardson et al., 2018b) and V2.0 (Seyednasrollah et al., 2019). Furthermore, details on image and time-series processing, data quality flags and filtering, and availability of interoperable software packages, such as phenocamr (Hufkens et al., 2018) and vegindex (<https://github.com/PhenoCamNetwork/python-vegindex>), can be found in these past data descriptors. No new software packages have been developed for this data release, and existing packages (e.g., phenocamr) do not yet support interfacing with cameraNDVI or the simplified data files.

  Separately, the distinction between Type I, II and III cameras is covered in a paragraph in the manuscript on L172-180.

- *It is tempting to ask for more of a discussion about how Gcc and cameraNDVI can be used beyond what the authors have mentioned in the last sentence of the discussion/manuscript. There are studies that have correlated Gcc and GPP for example, with highly variable results. As the authors of this study mention, NDVI can be more clearly*

*linked to LAI than Gcc. Thinking from an ecosystem modeling perspective it seems like the new cameraNDVI variable will be of greater benefit for evaluating LAI compared to using Gcc for either LAI or GPP. However, if models were to couple with a radiative transfer model then using GCC could be more directly linked to the models. I think it's beyond the scope to discuss this – and in any case there are plenty of other applications of these data. What I would suggest is if the authors think there are "good" and (perhaps especially) "bad" applications of either Gcc or cameraNDVI based on their expert knowledge, it would be a good opportunity to provide that perspective to the community. I personally would appreciate reading that. But again, I can see the argument that that is beyond the scope of this paper.*

**AC33:** thank you for this thoughtful comment. We agree it would be good to "step back" at the end of the discussion and provide more big picture thoughts on both strengths and caveats of using PhenoCam data, as well as highlighting how the standardized processing routines have led to significant enhancements in how phenology data can be used to study Earth systems. This last paragraph is on L598-617 and a copy of the text can be found in **AC12**.

- *Other things to potentially include that would be of benefit to the reader (especially point 1 for those that may be new to using PhenoCam data when reading this v3 paper):*

  - *A brief update to the software applications that can be used, especially with the new variables (or just a mention to see Seyednasrollah et al. (2019) if nothing has changed.*

    **AC34:** There are no new software packages to include, but we now make a general reference to a list of items readers should refer to in previous data descriptors on L263-269, and a copy of this text can be found in **AC32**.

  - *Seyednasrollah et al. (2019) did a comparison of transition dates between V1 and V2. This doesn't need to be repeated here, but a mention of the fact that the results are similar could be beneficial. I assume this is the case.*

    **AC35:** We appreciate this comment and agree this was overlooked when preparing V3.0. We have now conducted an analysis comparing transition dates between V3.0 and V2.0. For this analysis we found that transition dates from V3.0 very closely align to V2.0. We comment on this in the text (L467-472) which reads as follows: Finally, to ensure our data processing algorithm is consistent between versioned datasets, we compared transition dates in V3.0 to those in V2.0, similar to methods described in (Seyednasrollah et al., 2019). We found strong consistency between datasets, with r2 values > 98% and mean absolute errors (MAE) < 2.0 days. There were a small number of individual transition dates (~1%) between versions that we were unable to align for comparison; this primarily affected sites where ROIs have changed (e.g., FOV mask or time period differs), or in systems where the seasonal amplitude in GCC or the timing or number of seasonal transitions is more variable, such as in arid grasslands or in agricultural sites.

- *Will you publish the scripts used to process the phenocam data should anyone wish to look at that processing workflow?*

  **AC36:** These scripts for data processing are mentioned in the V1.0 data descriptor (Richardson et al. 2018b). We now reference this in the text on L263-269, and a copy of this new text can be found in **AC32**.

*References*

*Wingate, Lisa, Jérôme Ogée, Edoardo Cremonese, Gianluca Filippa, Toshie Mizunuma, Mirco Migliavacca, Christophe Moisy et al. "Interpreting canopy development and physiology using a European phenology camera network at flux sites." Biogeosciences 12, no. 20 (2015): 5995-6015.*

*Yan, D., Scott, R. L., Moore, D. J. P., Biederman, J. A., & Smith, W. K. (2019). Understanding the relationship between vegetation greenness and productivity across dryland ecosystems through the integration of PhenoCam, satellite, and eddy covariance data. Remote sensing of environment, 223, 50-62.*